# SNEkhorn: Dimension Reduction with Symmetric Entropic Affinities

**Hugues Van Assel**
ENS de Lyon, CNRS
UMPA UMR 5669
hugues.van_assel@ens-lyon.fr

**Titouan Vayer**
Univ. Lyon, ENS de Lyon, UCBL, CNRS, Inria
LIP UMR 5668
titouan.vayer@inria.fr

**Rémi Flamary**
École polytechnique, IP Paris, CNRS
CMAP UMR 7641
remi.flamary@polytechnique.edu

**Nicolas Courty**
Université Bretagne Sud, CNRS
IRISA UMR 6074
nicolas.courty@irisa.fr

## Abstract

Many approaches in machine learning rely on a weighted graph to encode the similarities between samples in a dataset. Entropic affinities (EAs), which are notably used in the popular Dimensionality Reduction (DR) algorithm t-SNE, are particular instances of such graphs. To ensure robustness to heterogeneous sampling densities, EAs assign a kernel bandwidth parameter to every sample in such a way that the entropy of each row in the affinity matrix is kept constant at a specific value, whose exponential is known as perplexity. EAs are inherently asymmetric and row-wise stochastic, but they are used in DR approaches after undergoing heuristic symmetrization methods that violate both the row-wise constant entropy and stochasticity properties. In this work, we uncover a novel characterization of EA as an optimal transport problem, allowing a natural symmetrization that can be computed efficiently using dual ascent. The corresponding novel affinity matrix derives advantages from symmetric doubly stochastic normalization in terms of clustering performance, while also effectively controlling the entropy of each row thus making it particularly robust to varying noise levels. Following, we present a new DR algorithm, SNEkhorn, that leverages this new affinity matrix. We show its clear superiority to existing approaches with several indicators on both synthetic and real-world datasets.

## 1 Introduction

Exploring and analyzing high-dimensional data is a core problem of data science that requires building low-dimensional and interpretable representations of the data through dimensionality reduction (DR). Ideally, these representations should preserve the data structure by mimicking, in the reduced representation space (called *latent space*), a notion of similarity between samples. We call *affinity* the weight matrix of a graph that encodes this similarity. It has positive entries and the higher the weight in position $(i, j)$, the higher the similarity or proximity between samples $i$ and $j$. Seminal approaches relying on affinities include Laplacian eigenmaps [2], spectral clustering [50] and semi-supervised learning [55]. Numerous methods can be employed to construct such affinities. A common choice is to use a kernel (*e.g.*, Gaussian) derived from a distance matrix normalized by a bandwidth parameter that usually has a large influence on the outcome of the algorithm. Indeed, excessively small kernel

37th Conference on Neural Information Processing Systems (NeurIPS 2023).

bandwidth can result in solely capturing the positions of closest neighbors, at the expense of large-scale dependencies. Inversely, setting too large a bandwidth blurs information about close-range pairwise relations. Ideally, one should select a different bandwidth for each point to accommodate varying sampling densities and noise levels. One approach is to compute the bandwidth of a point based on the distance from its $k$-th nearest neighbor [53]. However, this method fails to consider the entire distribution of distances. In general, selecting appropriate kernel bandwidths can be a laborious task, and many practitioners resort to greedy search methods. This can be limiting in some settings, particularly when dealing with large sample sizes.

**Entropic Affinities and SNE/t-SNE.** Entropic affinities (EAs) were first introduced in the seminal paper *Stochastic Neighbor Embedding* (SNE) [16]. It consists in normalizing each row $i$ of a distance matrix by a bandwidth parameter $\varepsilon_i$ such that the distribution associated with each row of the corresponding stochastic (*i.e.*, row-normalized) Gaussian affinity has a fixed entropy. The value of this entropy, whose exponential is called the *perplexity*, is then the only hyperparameter left to tune and has an intuitive interpretation as the number of effective neighbors of each point [49]. EAs are notoriously used to encode pairwise relations in a high-dimensional space for the DR algorithm t-SNE [46], among other DR methods including [6]. t-SNE is increasingly popular in many applied fields [20, 32] mostly due to its ability to represent clusters in the data [27, 5]. Nonetheless, one major flaw of EAs is that they are inherently directed and often require post-processing symmetrization.

**Doubly Stochastic Affinities.** Doubly stochastic (DS) affinities are non-negative matrices whose rows and columns have unit $\ell_1$ norm. In many applications, it has been demonstrated that DS affinity normalization (*i.e.*, determining the nearest DS matrix to a given affinity matrix) offers numerous benefits. First, it can be seen as a relaxation of k-means [51] and it is well-established that it enhances spectral clustering performances [10, 52, 1]. Additionally, DS matrices present the benefit of being invariant to the various Laplacian normalizations [50]. Recent observations indicate that the DS projection of the Gaussian kernel under the KL geometry is more resilient to heteroscedastic noise compared to its stochastic counterpart [23]. It also offers a more natural analog to the heat kernel [30]. These properties have led to a growing interest in DS affinities, with their use expanding to various applications such as smoothing filters [33], subspace clustering [25] and transformers [41].

**Contributions.** In this work, we study the missing link between EAs, which are easy to tune and adaptable to data with heterogeneous density, and DS affinities which have interesting properties in practical applications as aforementioned. Our main contributions are as follows. We uncover the convex optimization problem that underpins classical entropic affinities, exhibiting novel links with entropy-regularized Optimal Transport (OT) (Section 3.1). We then propose in Section 3.2 a principled symmetrization of entropic affinities. The latter enables controlling the entropy in each point, unlike t-SNE's post-processing symmetrization, and produces a genuinely doubly stochastic affinity. We show how to compute this new affinity efficiently using a dual ascent algorithm. In Section 4, we introduce SNEkhorn: a DR algorithm that couples this new symmetric entropic affinity with a doubly stochastic kernel in the low-dimensional embedding space, without sphere concentration issue [29]. We finally showcase the benefits of symmetric entropic affinities on a variety of applications in Section 5 including spectral clustering and DR experiments on datasets ranging from images to genomics data.

**Notations.** $[\![n]\!]$ denotes the set $\{1, ..., n\}$. $\exp$ and $\log$ applied to vectors/matrices are taken element-wise. $\mathbf{1} = (1, ..., 1)^\top$ is the vector of 1. $\langle \cdot, \cdot \rangle$ is the standard inner product for matrices/vectors. $\mathcal{S}$ is the space of $n \times n$ symmetric matrices. $\mathbf{P}_{i:}$ denotes the $i$-th row of a matrix $\mathbf{P}$. $\odot$ (*resp.* $\oslash$) stands for element-wise multiplication (*resp.* division) between vectors/matrices. For $\boldsymbol{\alpha}, \boldsymbol{\beta} \in \mathbb{R}^n, \boldsymbol{\alpha} \oplus \boldsymbol{\beta} \in \mathbb{R}^{n \times n}$ is $(\alpha_i + \beta_j)_{ij}$. The entropy of $\mathbf{p} \in \mathbb{R}_+^n$ is[1] $\mathrm{H}(\mathbf{p}) = -\sum_i p_i(\log(p_i) - 1) = -\langle \mathbf{p}, \log \mathbf{p} - \mathbf{1} \rangle$. The Kullback-Leibler divergence between two matrices $\mathbf{P}, \mathbf{Q}$ with nonnegative entries such that $Q_{ij} = 0 \implies P_{ij} = 0$ is $\mathrm{KL}(\mathbf{P}|\mathbf{Q}) = \sum_{ij} P_{ij} \left( \log(\frac{P_{ij}}{Q_{ij}}) - 1 \right) = \langle \mathbf{P}, \log(\mathbf{P} \oslash \mathbf{Q}) - \mathbf{1}\mathbf{1}^\top \rangle$.

## 2 Entropic Affinities, Dimensionality Reduction and Optimal Transport

Given a dataset $\mathbf{X} \in \mathbb{R}^{n \times p}$ of $n$ samples in dimension $p$, most DR algorithms compute a representation of $\mathbf{X}$ in a lower-dimensional latent space $\mathbf{Z} \in \mathbb{R}^{n \times q}$ with $q \ll p$ that faithfully captures and represents pairwise dependencies between the samples (or rows) in $\mathbf{X}$. This is generally achieved

---

[1]With the convention $0 \log 0 = 0$.

by optimizing $\mathbf{Z}$ such that the corresponding affinity matrix matches another affinity matrix defined from $\mathbf{X}$. These affinities are constructed from a matrix $\mathbf{C} \in \mathbb{R}^{n \times n}$ that encodes a notion of "distance" between the samples, *e.g.*, the squared Euclidean distance $C_{ij} = \|\mathbf{X}_{i:} - \mathbf{X}_{j:}\|_2^2$ or more generally any *cost matrix* $\mathbf{C} \in \mathcal{D} := \{\mathbf{C} \in \mathbb{R}_+^{n \times n} : \mathbf{C} = \mathbf{C}^\top \text{ and } C_{ij} = 0 \iff i = j\}$. A commonly used option is the Gaussian affinity that is obtained by performing row-wise normalization of the kernel $\exp(-\mathbf{C}/\varepsilon)$, where $\varepsilon > 0$ is the bandwidth parameter.

**Entropic Affinities (EAs).** Another frequently used approach to generate affinities from $\mathbf{C} \in \mathcal{D}$ is to employ *entropic affinities* [16]. The main idea is to consider *adaptive* kernel bandwidths $(\varepsilon_i^\star)_{i \in [\![n]\!]}$ to capture finer structures in the data compared to constant bandwidths [47]. Indeed, EAs rescale distances to account for the varying density across regions of the dataset. Given $\xi \in [\![n-1]\!]$, the goal of EAs is to build a Gaussian Markov chain transition matrix $\mathbf{P}^{\mathrm{e}}$ with prescribed entropy as

$$\forall i, \forall j, \ P_{ij}^{\mathrm{e}} = \frac{\exp\left(-C_{ij}/\varepsilon_i^\star\right)}{\sum_\ell \exp\left(-C_{i\ell}/\varepsilon_i^\star\right)} \tag{EA}$$
$$\text{with } \varepsilon_i^\star \in \mathbb{R}_+^* \text{ s.t. } \mathrm{H}(\mathbf{P}_{i:}^{\mathrm{e}}) = \log \xi + 1 \,.$$

The hyperparameter $\xi$, which is also known as *perplexity*, can be interpreted as the effective number of neighbors for each data point [49]. Indeed, a perplexity of $\xi$ means that each row of $\mathbf{P}^{\mathrm{e}}$ (which is a discrete probability since $\mathbf{P}^{\mathrm{e}}$ is row-wise stochastic) has the same entropy as a uniform distribution over $\xi$ neighbors. Therefore, it provides the practitioner with an interpretable parameter specifying which scale of dependencies the affinity matrix should faithfully capture. In practice, a root-finding algorithm is used to find the bandwidth parameters $(\varepsilon_i^\star)_{i \in [\![n]\!]}$ that satisfy the constraints [49]. Hereafter, with a slight abuse of language, we call $e^{\mathrm{H}(\mathbf{P}_{i:})-1}$ the perplexity of the point $i$.

**Dimension Reduction with SNE/t-SNE.** One of the main applications of EAs is the DR algorithm SNE [16]. We denote by $\mathbf{C_X} = \left(\|\mathbf{X}_{i:} - \mathbf{X}_{j:}\|_2^2\right)_{ij}$ and $\mathbf{C_Z} = \left(\|\mathbf{Z}_{i:} - \mathbf{Z}_{j:}\|_2^2\right)_{ij}$ the cost matrices derived from the rows (*i.e.*, the samples) of $\mathbf{X}$ and $\mathbf{Z}$ respectively. SNE focuses on minimizing in the latent coordinates $\mathbf{Z} \in \mathbb{R}^{n \times q}$ the objective $\mathrm{KL}(\mathbf{P}^{\mathrm{e}}|\mathbf{Q_Z})$ where $\mathbf{P}^{\mathrm{e}}$ solves (EA) with cost $\mathbf{C_X}$ and $[\mathbf{Q_Z}]_{ij} = \exp(-[\mathbf{C_Z}]_{ij})/(\sum_\ell \exp(-[\mathbf{C_Z}]_{i\ell}))$. In the seminal paper [46], a newer proposal for a *symmetric* version was presented, which has since replaced SNE in practical applications. Given a symmetric normalization for the similarities in latent space $[\widetilde{\mathbf{Q}}_{\mathbf{Z}}]_{ij} = \exp(-[\mathbf{C_Z}]_{ij})/\sum_{\ell,t} \exp(-[\mathbf{C_Z}]_{\ell t})$ it consists in solving

$$\min_{\mathbf{Z} \in \mathbb{R}^{n \times q}} \ \mathrm{KL}(\overline{\mathbf{P}^{\mathrm{e}}}|\widetilde{\mathbf{Q}}_{\mathbf{Z}}) \quad \text{where} \quad \overline{\mathbf{P}^{\mathrm{e}}} = \frac{1}{2}(\mathbf{P}^{\mathrm{e}} + \mathbf{P}^{\mathrm{e}\top}). \tag{Symmetric-SNE}$$

In other words, the affinity matrix $\overline{\mathbf{P}^{\mathrm{e}}}$ is the Euclidean projection of $\mathbf{P}^{\mathrm{e}}$ on the space of symmetric matrices $\mathcal{S}$: $\overline{\mathbf{P}^{\mathrm{e}}} = \mathrm{Proj}_{\mathcal{S}}^{\ell_2}(\mathbf{P}^{\mathrm{e}}) = \arg\min_{\mathbf{P} \in \mathcal{S}} \|\mathbf{P} - \mathbf{P}^{\mathrm{e}}\|_2$ (see Appendix A.1). Instead of the Gaussian kernel, the popular extension t-SNE [46] considers a different distribution in the latent space $[\widetilde{\mathbf{Q}}_{\mathbf{Z}}]_{ij} = (1 + [\mathbf{C_Z}]_{ij})^{-1}/\sum_{\ell,t}(1 + [\mathbf{C_Z}]_{\ell t})^{-1}$. In this formulation, $\widetilde{\mathbf{Q}}_{\mathbf{Z}}$ is a joint Student $t$-distribution that accounts for crowding effects: a relatively small distance in a high-dimensional space can be accurately represented by a significantly greater distance in the low-dimensional space.

Considering symmetric similarities is appealing since the proximity between two points is inherently symmetric. Nonetheless, the Euclidean projection in (Symmetric-SNE) *does not preserve the construction of entropic affinities*. In particular, $\overline{\mathbf{P}^{\mathrm{e}}}$ is not stochastic in general and $\mathrm{H}(\overline{\mathbf{P}}_{i:}^{\mathrm{e}}) \neq (\log \xi + 1)$ thus the entropy associated with each point is no longer controlled after symmetrization (see the bottom left plot of Figure 1). This is arguably one of the main drawbacks of the approach. By contrast, the $\mathbf{P}^{\mathrm{se}}$ affinity that will be introduced in Section 3 can accurately set the entropy in each point to the desired value $\log \xi + 1$. As shown in Figure 1 this leads to more faithful embeddings with better separation of the classes when combined with the t-SNEkhorn algorithm (Section 4).

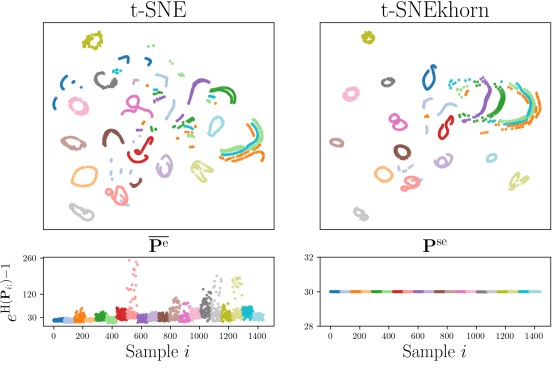

Figure 1: Top: COIL [34] embeddings with silhouette scores produced by t-SNE and t-SNEkhorn (our method introduced in Section 4) for $\xi = 30$. Bottom: $e^{\mathrm{H}(\mathbf{P}_{i:})-1}$ (*perplexity*) for each point $i$.

**Symmetric Entropy-Constrained Optimal Transport.** Entropy-regularized OT [37] and its connection to affinity matrices are crucial components in our solution. In the special case of uniform marginals, and for $\nu > 0$, entropic OT computes the minimum of $\mathbf{P} \mapsto \langle \mathbf{P}, \mathbf{C} \rangle - \nu \sum_i \mathrm{H}(\mathbf{P}_{i:})$ over the space of doubly stochastic matrices $\{\mathbf{P} \in \mathbb{R}_+^{n \times n} : \mathbf{P}\mathbf{1} = \mathbf{P}^\top \mathbf{1} = \mathbf{1}\}$. The optimal solution is the *unique* doubly stochastic matrix $\mathbf{P}^{\mathrm{ds}}$ of the form $\mathbf{P}^{\mathrm{ds}} = \mathrm{diag}(\mathbf{u})\mathbf{K}\,\mathrm{diag}(\mathbf{v})$ where $\mathbf{K} = \exp(-\mathbf{C}/\nu)$ is the Gibbs energy derived from $\mathbf{C}$ and $\mathbf{u}, \mathbf{v}$ are positive vectors that can be found with the celebrated Sinkhorn-Knopp's algorithm [8, 42]. Interestingly, when the cost $\mathbf{C}$ is *symmetric* (*e.g.*, $\mathbf{C} \in \mathcal{D}$) we can take $\mathbf{u} = \mathbf{v}$ [17, Section 5.2] so that the unique optimal solution is itself symmetric and writes

$$\mathbf{P}^{\mathrm{ds}} = \exp\left((\mathbf{f} \oplus \mathbf{f} - \mathbf{C})/\nu\right) \text{ where } \mathbf{f} \in \mathbb{R}^n \,. \tag{DS}$$

In this case, by relying on convex duality as detailed in Appendix A.2, an equivalent formulation for the symmetric entropic OT problem is

$$\min_{\mathbf{P} \in \mathbb{R}_+^{n \times n}} \quad \langle \mathbf{P}, \mathbf{C} \rangle \quad \text{s.t.} \quad \mathbf{P}\mathbf{1} = \mathbf{1}, \ \mathbf{P} = \mathbf{P}^\top \text{ and } \sum_i \mathrm{H}(\mathbf{P}_{i:}) \geq \eta \,, \tag{EOT}$$

where $0 \leq \eta \leq n(\log n + 1)$ is a constraint on the global entropy $\sum_i \mathrm{H}(\mathbf{P}_{i:})$ of the OT plan $\mathbf{P}$ which happens to be saturated at optimum (Appendix A.2). This constrained formulation of symmetric entropic OT will provide new insights into entropic affinities, as detailed in the next sections.

## 3 Symmetric Entropic Affinities

In this section, we present our first major contribution: symmetric entropic affinities. We begin by providing a new perspective on EAs through the introduction of an equivalent convex problem.

### 3.1 Entropic Affinities as Entropic Optimal Transport

We introduce the following set of matrices with row-wise stochasticity and entropy constraints:

$$\mathcal{H}_\xi := \left\{\mathbf{P} \in \mathbb{R}_+^{n \times n} \text{ s.t. } \mathbf{P}\mathbf{1} = \mathbf{1} \text{ and } \forall i, \ \mathrm{H}(\mathbf{P}_{i:}) \geq \log \xi + 1\right\} \,. \tag{1}$$

This space is convex since $\mathbf{p} \in \mathbb{R}_+^n \mapsto \mathrm{H}(\mathbf{p})$ is concave, thus its superlevel set is convex. In contrast to the entropic constraints utilized in standard entropic optimal transport which set a lower-bound on the *global* entropy, as demonstrated in the formulation (EOT), $\mathcal{H}_\xi$ imposes a constraint on the entropy of *each row* of the matrix $\mathbf{P}$. Our first contribution is to prove that EAs can be computed by solving a specific problem involving $\mathcal{H}_\xi$ (see Appendix A for the proof).

**Proposition 1.** *Let $\mathbf{C} \in \mathbb{R}^{n \times n}$ without constant rows. Then $\mathbf{P}^{\mathrm{e}}$ solves the entropic affinity problem (EA) with cost $\mathbf{C}$ if and only if $\mathbf{P}^{\mathrm{e}}$ is the unique solution of the convex problem*

$$\min_{\mathbf{P} \in \mathcal{H}_\xi} \langle \mathbf{P}, \mathbf{C} \rangle. \tag{EA as OT}$$

Interestingly, this result shows that EAs boil down to minimizing a transport objective with cost $\mathbf{C}$ and row-wise entropy constraints $\mathcal{H}_\xi$ where $\xi$ is the desired perplexity. As such, (EA as OT) can be seen as a specific *semi-relaxed* OT problem [39, 14] (*i.e.*, without the second constraint on the marginal $\mathbf{P}^\top \mathbf{1} = \mathbf{1}$) but with entropic constraints on the rows of $\mathbf{P}$. We also show that the optimal solution $\mathbf{P}^\star$ of (EA as OT) has *saturated entropy i.e.*, $\forall i, \mathrm{H}(\mathbf{P}_{i:}^\star) = \log \xi + 1$. In other words, relaxing the equality constraint in (EA) as an inequality constraint in $\mathbf{P} \in \mathcal{H}_\xi$ does not affect the solution while it allows reformulating entropic affinity as a convex optimization problem. To the best of our knowledge, this connection between OT and entropic affinities is novel and is an essential key to the method proposed in the next section.

**Remark 2.** The kernel bandwidth parameter $\varepsilon$ from the original formulation of entropic affinities (EA) is the Lagrange dual variable associated with the entropy constraint in (EA as OT). Hence computing $\varepsilon^\star$ in (EA) exactly corresponds to solving the dual problem of (EA as OT).

**Remark 3.** Let $\mathbf{K}_\sigma = \exp(-\mathbf{C}/\sigma)$. As shown in Appendix A.5, if $\varepsilon^\star$ solves (EA) and $\sigma \leq \min(\varepsilon^\star)$, then $\mathbf{P}^{\mathrm{e}} = \mathrm{Proj}_{\mathcal{H}_\xi}^{\mathrm{KL}}(\mathbf{K}_\sigma) = \arg\min_{\mathbf{P} \in \mathcal{H}_\xi} \mathrm{KL}(\mathbf{P}|\mathbf{K}_\sigma)$. Therefore $\mathbf{P}^{\mathrm{e}}$ can be seen as a KL Bregman projection [3] of a Gaussian kernel onto $\mathcal{H}_\xi$. Hence the input matrix in (Symmetric-SNE) is $\overline{\mathbf{P}^{\mathrm{e}}} = \mathrm{Proj}_{\mathcal{S}}^{\ell_2}(\mathrm{Proj}_{\mathcal{H}_\xi}^{\mathrm{KL}}(\mathbf{K}_\sigma))$ which corresponds to a surprising mixture of KL and orthogonal projections.

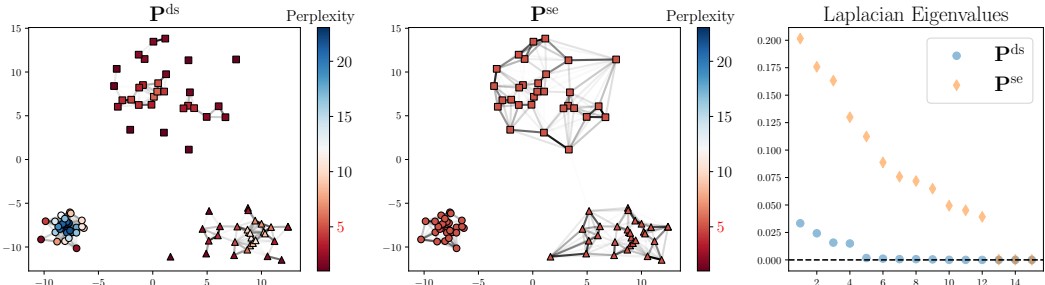

Figure 2: Samples from a mixture of three Gaussians with varying standard deviations. The edges' strength is proportional to the weights in the affinities $\mathbf{P}^{\mathrm{ds}}$ (DS) and $\mathbf{P}^{\mathrm{se}}$ (SEA) computed with $\xi = 5$ (for $\mathbf{P}^{\mathrm{ds}}$, $\xi$ is the average perplexity such that $\sum_i \mathrm{H}(\mathbf{P}^{\mathrm{ds}}_{i:}) = \sum_i \mathrm{H}(\mathbf{P}^{\mathrm{se}}_{i:})$). Points' color represents the perplexity $e^{\mathrm{H}(\mathbf{P}_{i:})-1}$. Right plot: smallest eigenvalues of the Laplacian for the two affinities.

## 3.2 Symmetric Entropic Affinity Formulation

Based on the previous formulation we now propose symmetric entropic affinities: a symmetric version of EAs that enables keeping the entropy associated with each row (or equivalently column) to the desired value of $\log \xi + 1$ while producing a symmetric doubly stochastic affinity matrix. Our strategy is to enforce symmetry through an additional constraint in (EA as OT), in a similar fashion as (EOT). More precisely we consider the convex optimization problem

$$\min_{\mathbf{P} \in \mathcal{H}_\xi \cap \mathcal{S}} \langle \mathbf{P}, \mathbf{C} \rangle .$$  (SEA)

where we recall that $\mathcal{S}$ is the set of $n \times n$ symmetric matrices. Note that for any $\xi \leq n - 1$, $\frac{1}{n} \mathbf{1} \mathbf{1}^\top \in \mathcal{H}_\xi \cap \mathcal{S}$ hence the set $\mathcal{H}_\xi \cap \mathcal{S}$ is a non-empty and convex set. We first detail some important properties of problem (SEA) (the proofs of the following results can be found in Appendix A.4).

**Proposition 4** (Saturation of the entropies). *Let $\mathbf{C} \in \mathcal{S}$ with zero diagonal, then* (SEA) *with cost $\mathbf{C}$ has a* unique solution *that we denote by $\mathbf{P}^{\mathrm{se}}$. If moreover $\mathbf{C} \in \mathcal{D}$, then for at least $n - 1$ indices $i \in [\![n]\!]$ the solution satisfies $\mathrm{H}(\mathbf{P}^{\mathrm{se}}_{i:}) = \log \xi + 1$.*

In other words, the unique solution $\mathbf{P}^{\mathrm{se}}$ has at least $n - 1$ saturated entropies *i.e.*, the corresponding $n - 1$ points have exactly a perplexity of $\xi$. In practice, with the algorithmic solution detailed below, we have observed that all $n$ entropies are saturated. Therefore, we believe that this proposition can be extended with a few more assumptions on $\mathbf{C}$. Accordingly, problem (SEA) allows accurate control over the point-wise entropies while providing a symmetric doubly stochastic matrix, unlike $\overline{\mathbf{P}^{\mathrm{e}}}$ defined in (Symmetric-SNE), as summarized in Table 1. In the sequel, we denote by $\mathrm{H}_{\mathrm{r}}(\mathbf{P}) = (\mathrm{H}(\mathbf{P}_{i:}))_i$ the vector of row-wise entropies of $\mathbf{P}$. We rely on the following result to compute $\mathbf{P}^{\mathrm{se}}$.

**Proposition 5** (Solving for SEA). *Let $\mathbf{C} \in \mathcal{D}, \mathcal{L}(\mathbf{P}, \boldsymbol{\gamma}, \boldsymbol{\lambda}) = \langle \mathbf{P}, \mathbf{C} \rangle + \langle \boldsymbol{\gamma}, (\log \xi + 1)\mathbf{1} - \mathrm{H}_{\mathrm{r}}(\mathbf{P}) \rangle + \langle \boldsymbol{\lambda}, \mathbf{1} - \mathbf{P}\mathbf{1} \rangle$ and $q(\boldsymbol{\gamma}, \boldsymbol{\lambda}) = \min_{\mathbf{P} \in \mathbb{R}^{n \times n}_+ \cap \mathcal{S}} \mathcal{L}(\mathbf{P}, \boldsymbol{\gamma}, \boldsymbol{\lambda})$. Strong duality holds for* (SEA). *Moreover, let $\boldsymbol{\gamma}^\star, \boldsymbol{\lambda}^\star \in \mathrm{argmax}_{\boldsymbol{\gamma} \geq 0, \boldsymbol{\lambda}} q(\boldsymbol{\gamma}, \boldsymbol{\lambda})$ be the optimal dual variables respectively associated with the entropy and marginal constraints. Then, for at least $n - 1$ indices $i \in [\![n]\!], \gamma^\star_i > 0$. When $\forall i \in [\![n]\!], \gamma^\star_i > 0$ then $\mathrm{H}_{\mathrm{r}}(\mathbf{P}^{\mathrm{se}}) = (\log \xi + 1)\mathbf{1}$ and $\mathbf{P}^{\mathrm{se}}$ has the form*

$$\mathbf{P}^{\mathrm{se}} = \exp\left((\boldsymbol{\lambda}^\star \oplus \boldsymbol{\lambda}^\star - 2\mathbf{C}) \oslash (\boldsymbol{\gamma}^\star \oplus \boldsymbol{\gamma}^\star)\right) .$$  (2)

By defining the symmetric matrix $\mathbf{P}(\boldsymbol{\gamma}, \boldsymbol{\lambda}) = \exp\left((\boldsymbol{\lambda} \oplus \boldsymbol{\lambda} - 2\mathbf{C}) \oslash (\boldsymbol{\gamma} \oplus \boldsymbol{\gamma})\right)$, we prove that, when $\boldsymbol{\gamma} > 0, \min_{\mathbf{P} \in \mathcal{S}} \mathcal{L}(\mathbf{P}, \boldsymbol{\gamma}, \boldsymbol{\lambda})$ has a unique solution given by $\mathbf{P}(\boldsymbol{\gamma}, \boldsymbol{\lambda})$ which implies $q(\boldsymbol{\gamma}, \boldsymbol{\lambda}) = \mathcal{L}(\mathbf{P}(\boldsymbol{\gamma}, \boldsymbol{\lambda}), \boldsymbol{\gamma}, \boldsymbol{\lambda})$. Thus the proposition shows that when $\boldsymbol{\gamma}^\star > 0, \mathbf{P}^{\mathrm{se}} = \mathbf{P}(\boldsymbol{\gamma}^\star, \boldsymbol{\lambda}^\star)$ where $\boldsymbol{\gamma}^\star, \boldsymbol{\lambda}^\star$ solve the following convex problem (as maximization of a *concave* objective)

$$\max_{\boldsymbol{\gamma} > 0, \boldsymbol{\lambda}} \mathcal{L}(\mathbf{P}(\boldsymbol{\gamma}, \boldsymbol{\lambda}), \boldsymbol{\gamma}, \boldsymbol{\lambda}).$$  (Dual-SEA)

Consequently, to find $\mathbf{P}^{\text{se}}$ we solve the problem (Dual-SEA). Although the form of $\mathbf{P}^{\text{se}}$ presented in Proposition 5 is only valid when $\gamma^\star$ is positive and we have only proved it for $n-1$ indices, we emphasize that if (Dual-SEA) has a finite solution, then it is equal to $\mathbf{P}^{\text{se}}$. Indeed in this case the solution satisfies the KKT system associated with (SEA).

Table 1: Properties of $\mathbf{P}^{\text{e}}$, $\overline{\mathbf{P}^{\text{e}}}$, $\mathbf{P}^{\text{ds}}$ and $\mathbf{P}^{\text{se}}$

| AFFINITY MATRIX REFERENCE | $\mathbf{P}^{\text{e}}$ [16] | $\overline{\mathbf{P}^{\text{e}}}$ [46] | $\mathbf{P}^{\text{ds}}$ [29] | $\mathbf{P}^{\text{se}}$ (SEA) |
|---|---|---|---|---|
| $\mathbf{P} = \mathbf{P}^\top$ | ✗ | ✓ | ✓ | ✓ |
| $\mathbf{P1} = \mathbf{P}^\top\mathbf{1} = \mathbf{1}$ | ✗ | ✗ | ✓ | ✓ |
| $\mathrm{H_r}(\mathbf{P}) = (\log \xi + 1)\mathbf{1}$ | ✓ | ✗ | ✗ | ✓ |

**Numerical optimization.** The dual problem (Dual-SEA) is concave and can be solved with guarantees through a dual ascent approach with closed-form gradients (using *e.g.*, SGD, BFGS [28] or ADAM [18]). At each gradient step, one can compute the current estimate $\mathbf{P}(\gamma, \lambda)$ while the gradients of the loss *w.r.t.* $\gamma$ and $\lambda$ are given respectively by the constraints $(\log \xi + 1)\mathbf{1} - \mathrm{H_r}(\mathbf{P}(\gamma, \lambda))$ and $\mathbf{1} - \mathbf{P}(\gamma, \lambda)\mathbf{1}$ (see *e.g.*, [4, Proposition 6.1.1]). Concerning time complexity, each step can be performed with $\mathcal{O}(n^2)$ algebraic operations. From a practical perspective, we found that using a change of variable $\gamma \leftarrow \gamma^2$ and optimize $\gamma \in \mathbb{R}^n$ leads to enhanced numerical stability.

**Remark 6.** In the same spirit as Remark 3, one can express $\mathbf{P}^{\text{se}}$ as a KL projection of $\mathbf{K}_\sigma = \exp(-\mathbf{C}/\sigma)$. Indeed, we show in Appendix A.5 that if $0 < \sigma \leq \min_i \gamma_i^\star$, then $\mathbf{P}^{\text{se}} = \mathrm{Proj}_{\mathcal{H}_\xi \cap \mathcal{S}}^{\mathrm{KL}}(\mathbf{K}_\sigma)$.

**Comparison between $\mathbf{P}^{\text{ds}}$ and $\mathbf{P}^{\text{se}}$.** In Figure 2 we illustrate the ability of our proposed affinity $\mathbf{P}^{\text{se}}$ to adapt to varying noise levels. In the OT problem that we consider, each sample is given a mass of one that is distributed over its neighbors (including itself since self-loops are allowed). For each sample, we refer to the entropy of the distribution over its neighbors as the *spreading* of its mass. One can notice that for $\mathbf{P}^{\text{ds}}$ (DS) (OT problem with global entropy constraint (EOT)) , the samples do not spread their mass evenly depending on the density around them. On the contrary, the per-row entropy constraints of $\mathbf{P}^{\text{se}}$ force equal spreading among samples. This can have benefits, particularly for clustering, as illustrated in the rightmost plot, which shows the eigenvalues of the associated Laplacian matrices (recall that the number of connected components equals the dimension of the null space of its Laplacian [7]). As can be seen, $\mathbf{P}^{\text{ds}}$ results in many unwanted clusters, unlike $\mathbf{P}^{\text{se}}$, which is robust to varying noise levels (its Laplacian matrix has only 3 vanishing eigenvalues). We further illustrate this phenomenon on Figure 3 with varying noise levels.

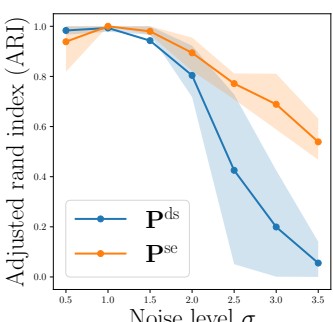

Figure 3: ARI spectral clustering on the example of three Gaussian clusters with variances: $\sigma^2$, $2\sigma^2$ and $3\sigma^2$ (as in Figure 2).

## 4  Optimal Transport for Dimension Reduction with SNEkhorn

In this section, we build upon symmetric entropic affinities to introduce SNEkhorn, a new DR algorithm that fully benefits from the advantages of doubly stochastic affinities.

**SNEkhorn's objective.** Our proposed method relies on doubly stochastic affinity matrices to capture the dependencies among the samples in both input *and* latent spaces. The KL divergence, which is the central criterion in most popular DR methods [44], is used to measure the discrepancy between the two affinities. As detailed in sections 2 and 3, $\mathbf{P}^{\text{se}}$ computed using the cost $[\mathbf{C_X}]_{ij} = \|\mathbf{X}_{i:} - \mathbf{X}_{j:}\|_2^2$, corrects for heterogeneity in the input data density by imposing point-wise entropy constraints. As we do not need such correction for embedding coordinates $\mathbf{Z}$ since they must be optimized, we opt for the standard affinity (DS) built as an OT transport plan with global entropy constraint (EOT). This OT plan can be efficiently computed using Sinkhorn's algorithm. More precisely, we propose the optimization problem

$$\min_{\mathbf{Z} \in \mathbb{R}^{n \times q}} \mathrm{KL}\big(\mathbf{P}^{\text{se}} | \mathbf{Q}_{\mathbf{Z}}^{\text{ds}}\big), \qquad \text{(SNEkhorn)}$$

where $\mathbf{Q}_{\mathbf{Z}}^{\text{ds}} = \exp(\mathbf{f_Z} \oplus \mathbf{f_Z} - \mathbf{C_Z})$ stands for the (DS) affinity computed with cost $[\mathbf{C_Z}]_{ij} = \|\mathbf{Z}_{i:} - \mathbf{Z}_{j:}\|_2^2$ and $\mathbf{f_Z}$ is the optimal dual variable found by Sinkhorn's algorithm. We set the bandwidth to $\nu = 1$ in $\mathbf{Q}_{\mathbf{Z}}^{\text{ds}}$ similarly to [46] as the bandwidth in the low dimensional space only affects the scales

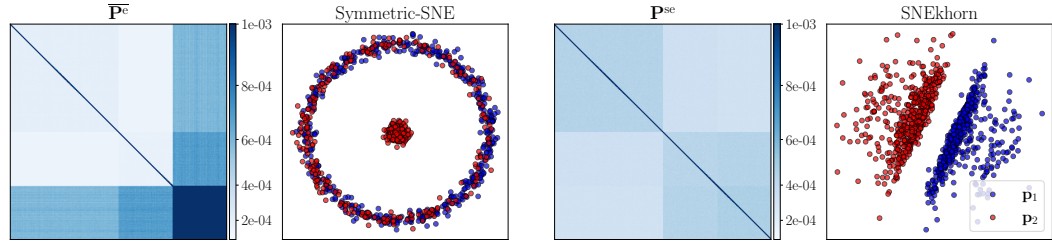

Figure 5: From left to right: entries of $\overline{\mathbf{P}^{\mathrm{e}}}$ (Symmetric-SNE) and associated embeddings generated using $\overline{\mathbf{P}^{\mathrm{e}}}$. Then $\mathbf{P}^{\mathrm{se}}$ (SEA) matrix and associated SNEkhorn embeddings. Perplexity $\xi = 30$.

of the embeddings and not their shape. Keeping only the terms that depend on $\mathbf{Z}$ and relying on the double stochasticity of $\mathbf{P}^{\mathrm{se}}$, the objective in (SNEkhorn) can be expressed as $\langle \mathbf{P}^{\mathrm{se}}, \mathbf{C}_\mathbf{Z} \rangle - 2\langle \mathbf{f}_\mathbf{Z}, \mathbf{1} \rangle$.

**Heavy-tailed kernel in latent space.** Since it is well known that heavy-tailed kernels can be beneficial in DR [21], we propose an extension called t-SNEkhorn that simply amounts to computing a doubly stochastic student-t kernel in the low-dimensional space. With our construction, it corresponds to choosing the cost $[\mathbf{C}_\mathbf{Z}]_{ij} = \log(1 + \|\mathbf{Z}_{i:} - \mathbf{Z}_{j:}\|_2^2)$ instead of $\|\mathbf{Z}_{i:} - \mathbf{Z}_{j:}\|_2^2$.

**Inference.** This new DR objective involves computing a doubly stochastic normalization for each update of $\mathbf{Z}$. Interestingly, to compute the optimal dual variable $\mathbf{f}_\mathbf{Z}$ in $\mathbf{Q}_\mathbf{Z}^{\mathrm{ds}}$, we leverage a well-conditioned Sinkhorn fixed point iteration [19, 13], which converges extremely fast in the symmetric setting:

$$\forall i, \; [\mathbf{f}_\mathbf{Z}]_i \leftarrow \frac{1}{2}\left([\mathbf{f}_\mathbf{Z}]_i - \log\sum_k \exp\left([\mathbf{f}_\mathbf{Z}]_k - [\mathbf{C}_\mathbf{Z}]_{ki}\right)\right) . \qquad \text{(Sinkhorn)}$$

On the right side of Figure 4, we plot $\|\mathbf{Q}_\mathbf{Z}^{\mathrm{ds}}\mathbf{1} - \mathbf{1}\|_\infty$ as a function of (Sinkhorn) iterations for a toy example presented in Section 5. In most practical cases, we found that about 10 iterations were enough to reach a sufficiently small error. $\mathbf{Z}$ is updated through gradient descent with gradients obtained by performing back-propagation through the Sinkhorn iterations. These iterations can be further accelerated with a *warm start* strategy by plugging the last $\mathbf{f}_\mathbf{Z}$ to initialize the current one.

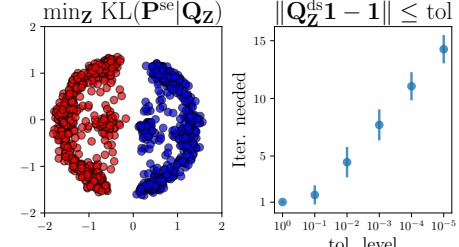

Figure 4: Left: SNEkhorn embedding on the simulated data of Section 5 using $\widetilde{\mathbf{Q}}_\mathbf{Z}$ instead of $\mathbf{Q}_\mathbf{Z}^{\mathrm{ds}}$ with $\xi = 30$. Right: number of iterations needed to achieve $\|\mathbf{Q}_\mathbf{Z}^{\mathrm{ds}}\mathbf{1} - \mathbf{1}\|_\infty \leq$ tol with (Sinkhorn).

**Related work.** Using doubly stochastic affinities for SNE has been proposed in [29], with two key differences from our work. First, they do not consider EAs and resort to $\mathbf{P}^{\mathrm{ds}}$ (DS). This affinity, unlike $\mathbf{P}^{\mathrm{se}}$, is not adaptive to the data heterogeneous density (as illusrated in Figure 2). Second, they use the affinity $\widetilde{\mathbf{Q}}_\mathbf{Z}$ in the low-dimensional space and illustrate empirically that matching the latter with a doubly stochastic matrix (*e.g.*, $\mathbf{P}^{\mathrm{ds}}$ or $\mathbf{P}^{\mathrm{se}}$) can sometimes impose spherical constraints on the embedding $\mathbf{Z}$. This is detrimental for projections onto a $2D$ flat space (typical use case of DR) where embeddings tend to form circles. This can be verified on the left side of Figure 4. In contrast, in SNEkhorn, the latent affinity *is also doubly stochastic* so that latent coordinates $\mathbf{Z}$ are not subject to spherical constraints anymore. The corresponding SNEkhorn embedding is shown in Figure 5 (bottom right).

## 5 Numerical experiments

This section aims to illustrate the performances of the proposed affinity matrix $\mathbf{P}^{\mathrm{se}}$ (SEA) and DR method SNEkhorn at faithfully representing dependencies and clusters in low dimensions. First, we showcase the relevance of our approach on a simple synthetic dataset with heteroscedastic noise. Then,

we evaluate the spectral clustering performances of symmetric entropic affinities before benchmarking t-SNEkhorn with t-SNE and UMAP [31] on real-world images and genomics datasets.[2]

**Simulated data.** We consider the toy dataset with heteroscedastic noise from [23]. It consists of sampling uniformly two vectors $\mathbf{p}_1$ and $\mathbf{p}_2$ in the $10^4$-dimensional probability simplex. $n = 10^3$ samples are then generated as $\mathbf{x}_i = \tilde{\mathbf{x}}_i/(\sum_j \tilde{x}_{ij})$ where

$$
\tilde{\mathbf{x}}_i \sim \left\{ \begin{array}{ll} \mathcal{M}(1000, \mathbf{p}_1), & 1 \le i \le 500 \\ \mathcal{M}(1000, \mathbf{p}_2), & 501 \le i \le 750 \\ \mathcal{M}(2000, \mathbf{p}_2), & 751 \le i \le 1000 \,. \end{array} \right.
$$

where $\mathcal{M}$ stands for the multinomial distribution. The goal of the task is to test the robustness to heteroscedastic noise. Indeed, points generated using $\mathbf{p}_2$ exhibit different levels of noise due to various numbers of multinomial trials to form an estimation of $\mathbf{p}_2$. This typically occurs in real-world scenarios when the same entity is measured using different experimental setups thus creating heterogeneous technical noise levels (*e.g.*, in single-cell sequencing [20]). This phenomenon is known as *batch effect* [43]. In Figure 5, we show that, unlike $\overline{\mathbf{P}^e}$ (Symmetric-SNE), $\mathbf{P}^{se}$ (SEA) manages to properly filter the noise (top row) to discriminate between samples generated by $\mathbf{p}_1$ and $\mathbf{p}_2$, and represent these two clusters separately in the embedding space (bottom row). In contrast, $\overline{\mathbf{P}^e}$ and SNE are misled by the batch effect. This shows that $\overline{\mathbf{P}^e}$ doesn't fully benefit from the adaptivity of EAs due to poor normalization and symmetrization. This phenomenon partly explains the superiority of SNEkhorn and t-SNEkhorn over current approaches on real-world datasets as illustrated below.

**Real-world datasets.** We then experiment with various labeled classification datasets including images and genomic data. For images, we use COIL 20 [34], OLIVETTI faces [12], UMNIST [15] and CIFAR 10 [22]. For CIFAR, we experiment with features obtained from the last hidden layer of a pre-trained ResNet [38] while for the other three datasets, we take as input the raw pixel data. Regarding genomics data, we consider the Curated Microarray Database (CuMiDa) [11] made of microarray datasets for various types of cancer, as well as the pre-processed SNAREseq (chromatin accessibility) and scGEM (gene expression)

Table 2: ARI ($\times 100$) clustering scores on genomics.

| DATA SET | $\overline{\mathbf{P}^{rs}}$ | $\mathbf{P}^{ds}$ | $\mathbf{P}^{st}$ | $\overline{\mathbf{P}^e}$ | $\mathbf{P}^{se}$ |
|---|---|---|---|---|---|
| LIVER (14520) | 75.8 | 75.8 | 84.9 | 80.8 | **85.9** |
| BREAST (70947) | **30.0** | **30.0** | 26.5 | 23.5 | 28.5 |
| LEUKEMIA (28497) | 43.7 | 44.1 | 49.7 | 42.5 | **50.6** |
| COLORECTAL (44076) | **95.9** | **95.9** | 93.9 | **95.9** | **95.9** |
| LIVER (76427) | 76.7 | 76.7 | **83.3** | 81.1 | 81.1 |
| BREAST (45827) | 43.6 | 53.8 | 74.7 | 71.5 | **77.0** |
| COLORECTAL (21510) | 57.6 | 57.6 | 54.7 | **94.0** | 79.3 |
| RENAL (53757) | 47.6 | 47.6 | **49.5** | **49.5** | **49.5** |
| PROSTATE (6919) | 12.0 | 13.0 | 13.2 | 16.3 | **17.4** |
| THROAT (42743) | 9.29 | 9.29 | 11.4 | 11.8 | **44.2** |
| scGEM | 57.3 | 58.5 | **74.8** | 69.9 | 71.6 |
| SNAREseq | 8.89 | 9.95 | 46.3 | 55.4 | **96.6** |

datasets used in [9]. For CuMiDa, we retain the datasets with most samples. For all the datasets, when the data dimension exceeds 50 we apply a pre-processing step of PCA in dimension 50, as usually done in practice [46]. In the following experiments, when not specified the hyperparameters are set to the value leading to the best average score on five different seeds with grid-search. For perplexity parameters, we test all multiples of 10 in the interval $[10, \min(n, 300)]$ where $n$ is the number of samples in the dataset. We use the same grid for the $k$ of the self-tuning affinity $\mathbf{P}^{st}$ [53] and for the `n_neighbors` parameter of UMAP. For scalar bandwidths, we consider powers of 10 such that the corresponding affinities' average perplexity belongs to the perplexity range.

**Spectral Clustering.** Building on the strong connections between spectral clustering mechanisms and t-SNE [44, 27] we first consider spectral clustering tasks to evaluate the affinity matrix $\mathbf{P}^{se}$ (SEA) and compare it against $\overline{\mathbf{P}^e}$ (Symmetric-SNE). We also consider two versions of the Gaussian affinity with scalar bandwidth $\mathbf{K} = \exp(-\mathbf{C}/\nu)$: the symmetrized row-stochastic $\overline{\mathbf{P}^{rs}} = \text{Proj}_{\mathcal{S}}^{\ell_2}(\mathbf{P}^{rs})$ where $\mathbf{P}^{rs}$ is $\mathbf{K}$ normalized by row and $\mathbf{P}^{ds}$ (DS). We also consider the adaptive Self-Tuning $\mathbf{P}^{st}$ affinity from [53] which relies on an adaptive bandwidth corresponding to the distance from the $k$-th nearest neighbor of each point. We use the spectral clustering implementation of `scikit-learn` [36] with default parameters which uses the unnormalized graph Laplacian. We measure the quality of clustering using the Adjusted Rand Index (ARI). Looking at both Table 2 and Figure 6, one can notice that, in general, symmetric entropic affinities yield better results than usual entropic affinities with significant improvements in some datasets (*e.g.*, throat microarray and SNAREseq). Overall

---

[2]Our code is available at `https://github.com/PythonOT/SNEkhorn`.

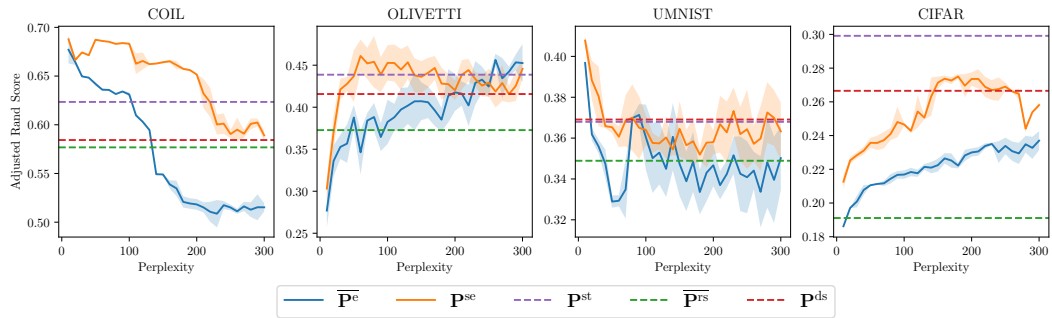

Figure 6: ARI spectral clustering score as a function of the perplexity parameter for image datasets.

Table 3: Scores for the UMAP, t-SNE and t-SNEkhorn embeddings.

| | Silhouette ($\times 100$) | | | Trustworthiness ($\times 100$) | | |
|---|---|---|---|---|---|---|
| | UMAP | t-SNE | t-SNEkhorn | UMAP | t-SNE | t-SNEkhorn |
| COIL | $20.4 \pm 3.3$ | $30.7 \pm 6.9$ | $\mathbf{52.3 \pm 1.1}$ | $99.6 \pm 0.1$ | $99.6 \pm 0.1$ | $\mathbf{99.9 \pm 0.1}$ |
| OLIVETTI | $6.4 \pm 4.2$ | $4.5 \pm 3.1$ | $\mathbf{15.7 \pm 2.2}$ | $96.5 \pm 1.3$ | $96.2 \pm 0.6$ | $\mathbf{98.0 \pm 0.4}$ |
| UMNIST | $-1.4 \pm 2.7$ | $-0.2 \pm 1.5$ | $\mathbf{25.4 \pm 4.9}$ | $93.0 \pm 0.4$ | $99.6 \pm 0.2$ | $\mathbf{99.8 \pm 0.1}$ |
| CIFAR | $13.6 \pm 2.4$ | $18.3 \pm 0.8$ | $\mathbf{31.5 \pm 1.3}$ | $90.2 \pm 0.8$ | $90.1 \pm 0.4$ | $\mathbf{92.4 \pm 0.3}$ |
| Liver (14520) | $49.7 \pm 1.3$ | $50.9 \pm 0.7$ | $\mathbf{61.1 \pm 0.3}$ | $89.2 \pm 0.7$ | $90.4 \pm 0.4$ | $\mathbf{92.3 \pm 0.3}$ |
| Breast (70947) | $28.6 \pm 0.8$ | $29.0 \pm 0.2$ | $\mathbf{31.2 \pm 0.2}$ | $90.9 \pm 0.5$ | $91.3 \pm 0.3$ | $\mathbf{93.2 \pm 0.4}$ |
| Leukemia (28497) | $22.3 \pm 0.7$ | $20.6 \pm 0.7$ | $\mathbf{26.2 \pm 2.3}$ | $90.4 \pm 1.1$ | $92.3 \pm 0.8$ | $\mathbf{94.3 \pm 0.5}$ |
| Colorectal (44076) | $67.6 \pm 2.2$ | $69.5 \pm 0.5$ | $\mathbf{74.8 \pm 0.4}$ | $93.2 \pm 0.7$ | $93.7 \pm 0.5$ | $\mathbf{94.3 \pm 0.6}$ |
| Liver (76427) | $39.4 \pm 4.3$ | $38.3 \pm 0.9$ | $\mathbf{51.2 \pm 2.5}$ | $85.9 \pm 0.4$ | $89.4 \pm 1.0$ | $\mathbf{92.0 \pm 1.0}$ |
| Breast (45827) | $35.4 \pm 3.3$ | $39.5 \pm 1.9$ | $\mathbf{44.4 \pm 0.5}$ | $93.2 \pm 0.4$ | $94.3 \pm 0.2$ | $\mathbf{94.7 \pm 0.3}$ |
| Colorectal (21510) | $38.0 \pm 1.3$ | $\mathbf{42.3 \pm 0.6}$ | $35.1 \pm 2.1$ | $85.6 \pm 0.7$ | $\mathbf{88.3 \pm 0.9}$ | $88.2 \pm 0.7$ |
| Renal (53757) | $44.4 \pm 1.5$ | $45.9 \pm 0.3$ | $\mathbf{47.8 \pm 0.1}$ | $93.9 \pm 0.2$ | $\mathbf{94.6 \pm 0.2}$ | $94.0 \pm 0.2$ |
| Prostate (6919) | $5.4 \pm 2.7$ | $8.1 \pm 0.2$ | $\mathbf{9.1 \pm 0.1}$ | $77.6 \pm 1.8$ | $\mathbf{80.6 \pm 0.2}$ | $73.1 \pm 0.5$ |
| Throat (42743) | $26.7 \pm 2.4$ | $28.0 \pm 0.3$ | $\mathbf{32.3 \pm 0.1}$ | $\mathbf{91.5 \pm 1.3}$ | $88.6 \pm 0.8$ | $86.8 \pm 1.0$ |
| scGEM | $26.9 \pm 3.7$ | $33.0 \pm 1.1$ | $\mathbf{39.3 \pm 0.7}$ | $95.0 \pm 1.3$ | $96.2 \pm 0.6$ | $\mathbf{96.8 \pm 0.3}$ |
| SNAREseq | $6.8 \pm 6.0$ | $35.8 \pm 5.2$ | $\mathbf{67.9 \pm 1.2}$ | $93.1 \pm 2.8$ | $99.1 \pm 0.1$ | $\mathbf{99.2 \pm 0.1}$ |

$\mathbf{P}^{\mathrm{se}}$ outperforms all the other affinities in 8 out of 12 datasets. This shows that the adaptivity of EAs is crucial. Figure 6 also shows that this superiority is verified for the whole range of perplexities. This can be attributed to the fact that symmetric entropic affinities combine the advantages of doubly stochastic normalization in terms of clustering and of EAs in terms of adaptivity. In the next experiment, we show that these advantages translate into better clustering and neighborhood retrieval at the embedding level when running SNEkhorn.

**Dimension Reduction.** To guarantee a fair comparison, we implemented not only SNEkhorn, but also t-SNE and UMAP in `PyTorch` [35]. Note that UMAP also relies on adaptive affinities but sets the degree of each node (related to the hyperparameter `n_neighbors` which plays a similar role to the perplexity) rather than the entropy. All models were optimized using `ADAM` [18] with default parameters and the same stopping criterion: the algorithm stops whenever the relative variation of the loss becomes smaller than $10^{-5}$. For each run, we draw independent

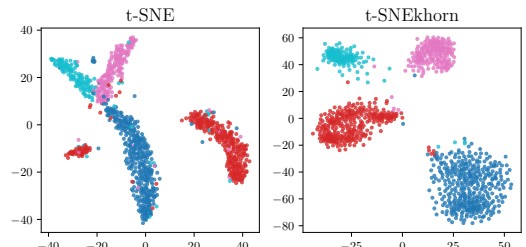

Figure 7: SNAREseq embeddings produced by t-SNE and t-SNEkhorn with $\xi = 50$.

$\mathcal{N}(0, 1)$ coordinates and use this same matrix to initialize all the methods that we wish to compare. To evaluate the embeddings' quality, we make use of the silhouette [40] and trustworthiness [48]

scores from `scikit-learn` [36] with default parameters. While the former relies on class labels, the latter measures the agreement between the neighborhoods in input and output spaces, thus giving two complementary metrics to properly evaluate the embeddings. The results, presented in Table 3, demonstrate the notable superiority of t-SNEkhorn compared to the commonly used t-SNE and UMAP algorithms. A sensitivity analysis on perplexity can also be found in Appendix B. Across the 16 datasets examined, t-SNEkhorn almost consistently outperformed the others, achieving the highest silhouette score on 15 datasets and the highest trustworthiness score on 12 datasets. To visually assess the quality of the embeddings, we provide SNAREseq embeddings in Figure 7. Notably, one can notice that the use of t-SNEkhorn results in improved class separation compared to t-SNE.

## 6 Conclusion

We have introduced a new principled and efficient method for constructing symmetric entropic affinities. Unlike the current formulation that enforces symmetry through an orthogonal projection, our approach allows control over the entropy in each point thus achieving entropic affinities' primary goal. Additionally, it produces a DS-normalized affinity and thus benefits from the well-known advantages of this normalization. Our affinity takes as input the same perplexity parameter as EAs and can thus be used with little hassle for practitioners. We demonstrate experimentally that both our affinity and DR algorithm (SNEkhorn), leveraging a doubly stochastic kernel in the latent space, achieve substantial improvements over existing approaches.

Note that in the present work, we do not address the issue of large-scale dependencies that are not faithfully represented in the low-dimensional space [44]. The latter shall be treated in future works. Among other promising research directions, one could focus on building multi-scale versions of symmetric entropic affinities [24] as well as fast approximations for SNEkhorn forces by adapting *e.g.*, Barnes-Hut [45] or interpolation-based methods [26] to the doubly stochastic setting. It could also be interesting to use SEAs in order to study the training dynamics of transformers [54].

## Acknowledgments

The authors are grateful to Mathurin Massias, Jean Feydy and Aurélien Garivier for insightful discussions. This project was supported in part by the ANR projects AllegroAssai ANR-19-CHIA-0009, SingleStatOmics ANR-18-CE45-0023 and OTTOPIA ANR-20-CHIA-0030. This work was also supported by the ACADEMICS grant of the IDEXLYON, project of the Université de Lyon, PIA operated by ANR-16-IDEX-0005.

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

# A Proofs

## A.1 Euclidean Projection onto $\mathcal{S}$

For the problem $\arg\min_{\mathbf{P}\in\mathcal{S}} \|\mathbf{P} - \mathbf{K}\|_2^2$, the Lagrangian takes the form, with $\mathbf{W} \in \mathbb{R}^{n\times n}$,

$$\mathcal{L}(\mathbf{P}, \mathbf{W}) = \|\mathbf{P} - \mathbf{K}\|_2^2 + \langle \mathbf{W}, \mathbf{P} - \mathbf{P}^\top \rangle. \tag{3}$$

Cancelling the gradient of $\mathcal{L}$ with respect to $\mathbf{P}$ gives $2(\mathbf{P}^\star - \mathbf{K}) + \mathbf{W} - \mathbf{W}^\top = \mathbf{0}$. Thus $\mathbf{P}^\star = \mathbf{K} + \frac{1}{2}\left(\mathbf{W}^\top - \mathbf{W}\right)$. Using the symmetry constraint on $\mathbf{P}^\star$ yields $\mathbf{P}^\star = \frac{1}{2}\left(\mathbf{K} + \mathbf{K}^\top\right)$. Hence we have:

$$\arg\min_{\mathbf{P}\in\mathcal{S}} \|\mathbf{P} - \mathbf{K}\|_2^2 = \frac{1}{2}\left(\mathbf{K} + \mathbf{K}^\top\right). \tag{4}$$

## A.2 From Symmetric Entropy-Constrained OT to Sinkhorn Iterations

In this section, we derive Sinkhorn iterations from the problem (EOT). Let $\mathbf{C} \in \mathcal{D}$. We start by making the constraints explicit.

$$\min_{\mathbf{P}\in\mathbb{R}_+^{n\times n}} \quad \langle \mathbf{P}, \mathbf{C}\rangle \tag{5}$$

$$\text{s.t.} \quad \sum_{i\in[\![n]\!]} \mathrm{H}(\mathbf{P}_{i:}) \geq \eta \tag{6}$$

$$\mathbf{P}\mathbf{1} = \mathbf{1}, \quad \mathbf{P} = \mathbf{P}^\top. \tag{7}$$

For the above convex problem the Lagrangian writes, where $\nu \in \mathbb{R}_+$, $\mathbf{f} \in \mathbb{R}^n$ and $\mathbf{\Gamma} \in \mathbb{R}^{n\times n}$:

$$\mathcal{L}(\mathbf{P}, \mathbf{f}, \nu, \mathbf{\Gamma}) = \langle \mathbf{P}, \mathbf{C}\rangle + \left\langle \nu, \eta - \sum_{i\in[\![n]\!]} \mathrm{H}(\mathbf{P}_i)\right\rangle + 2\langle \mathbf{f}, \mathbf{1} - \mathbf{P}\mathbf{1}\rangle + \langle \mathbf{\Gamma}, \mathbf{P} - \mathbf{P}^\top\rangle. \tag{8}$$

Strong duality holds and the first order KKT condition gives for the optimal primal $\mathbf{P}^\star$ and dual $(\nu^\star, \mathbf{f}^\star, \mathbf{\Gamma}^\star)$ variables:

$$\nabla_{\mathbf{P}}\mathcal{L}(\mathbf{P}^\star, \mathbf{f}^\star, \nu^\star, \mathbf{\Gamma}^\star) = \mathbf{C} + \nu^\star\log\mathbf{P}^\star - 2\mathbf{f}^\star\mathbf{1}^\top + \mathbf{\Gamma}^\star - \mathbf{\Gamma}^{\star\top} = \mathbf{0}. \tag{9}$$

Since $\mathbf{P}^\star, \mathbf{C} \in \mathcal{S}$ we have $\mathbf{\Gamma}^\star - \mathbf{\Gamma}^{\star\top} = \mathbf{f}^\star\mathbf{1}^\top - \mathbf{1}\mathbf{f}^{\star\top}$. Hence $\mathbf{C} + \nu^\star\log\mathbf{P}^\star - \mathbf{f}^\star \oplus \mathbf{f}^\star = \mathbf{0}$. Suppose that $\nu^\star = 0$ then the previous reasoning implies that $\forall(i,j), C_{ij} = f_i^\star + f_j^\star$. Using that $\mathbf{C} \in \mathcal{D}$ we have $C_{ii} = C_{jj} = 0$ thus $\forall i, f_i^\star = 0$ and thus this would imply that $\mathbf{C} = 0$ which is not allowed by hypothesis. Therefore $\nu^\star \neq 0$ and the entropy constraint is saturated at the optimum by complementary slackness. Isolating $\mathbf{P}^\star$ then yields:

$$\mathbf{P}^\star = \exp\left((\mathbf{f}^\star \oplus \mathbf{f}^\star - \mathbf{C})/\nu^\star\right). \tag{10}$$

$\mathbf{P}^\star$ must be primal feasible in particular $\mathbf{P}^\star\mathbf{1} = \mathbf{1}$. This constraint gives us the Sinkhorn fixed point relation for $\mathbf{f}^\star$:

$$\forall i \in [\![n]\!], \quad [\mathbf{f}^\star]_i = -\nu^\star\,\mathrm{LSE}\left((\mathbf{f}^\star - \mathbf{C}_{:i})/\nu^\star\right), \tag{11}$$

where for a vector $\boldsymbol{\alpha}$, we use the notation $\mathrm{LSE}(\boldsymbol{\alpha}) = \log\sum_k\exp(\alpha_k)$.

## A.3 Proof of Proposition 1

We recall the result

**Proposition 1.** *Let $\mathbf{C} \in \mathbb{R}^{n\times n}$ without constant rows. Then $\mathbf{P}^{\mathrm{e}}$ solves the entropic affinity problem (EA) with cost $\mathbf{C}$ if and only if $\mathbf{P}^{\mathrm{e}}$ is the unique solution of the convex problem*

$$\min_{\mathbf{P}\in\mathcal{H}_\xi} \langle\mathbf{P}, \mathbf{C}\rangle. \tag{EA as OT}$$

*Proof.* We begin by rewriting the above problem to make the constraints more explicit.

$$\min_{\mathbf{P}\in\mathbb{R}_+^{n\times n}} \quad \langle\mathbf{P}, \mathbf{C}\rangle$$

$$\text{s.t.} \quad \forall i,\, \mathrm{H}(\mathbf{P}_{i:}) \geq \log\xi + 1$$

$$\mathbf{P}\mathbf{1} = \mathbf{1}.$$

By concavity of entropy, one has that the entropy constraint is convex thus the above primal problem is a convex optimization problem. Moreover, the latter is strictly feasible for any $\xi \in [\![n-1]\!]$. Therefore Slater's condition is satisfied and strong duality holds.

Introducing the dual variables $\boldsymbol{\lambda} \in \mathbb{R}^n$ and $\boldsymbol{\varepsilon} \in \mathbb{R}_+^n$, the Lagrangian of the above problem writes:

$$\mathcal{L}(\mathbf{P}, \boldsymbol{\lambda}, \boldsymbol{\varepsilon}) = \langle \mathbf{P}, \mathbf{C} \rangle + \langle \boldsymbol{\varepsilon}, (\log \xi + 1)\mathbf{1} - \mathrm{H_r}(\mathbf{P}) \rangle + \langle \boldsymbol{\lambda}, \mathbf{1} - \mathbf{P}\mathbf{1} \rangle, \tag{12}$$

where we recall that $\mathrm{H_r}(\mathbf{P}) = (\mathrm{H}(\mathbf{P}_{i:}))_i$. Note that we will deal with the constraint $\mathbf{P} \in \mathbb{R}_+^{n \times n}$ directly, hence there is no associated dual variable. Since strong duality holds, for any solution $\mathbf{P}^\star$ to the primal problem and any solution $(\boldsymbol{\varepsilon}^\star, \boldsymbol{\lambda}^\star)$ to the dual problem, the pair $\mathbf{P}^\star, (\boldsymbol{\varepsilon}^\star, \boldsymbol{\lambda}^\star)$ must satisfy the Karush-Kuhn-Tucker (KKT) conditions. The first-order optimality condition gives:

$$\nabla_{\mathbf{P}} \mathcal{L}(\mathbf{P}^\star, \boldsymbol{\varepsilon}^\star, \boldsymbol{\lambda}^\star) = \mathbf{C} + \mathrm{diag}(\boldsymbol{\varepsilon}^\star) \log \mathbf{P}^\star - \boldsymbol{\lambda}^\star \mathbf{1}^\top = \mathbf{0}. \tag{first-order}$$

Assume that there exists $\ell \in [\![n]\!]$ such that $\varepsilon_\ell^\star = 0$. Then (first-order) gives that the $\ell^{th}$ row of $\mathbf{C}$ is constant which is not allowed by hypothesis. Therefore $\boldsymbol{\varepsilon}^\star > \mathbf{0}$ (*i.e.*, $\boldsymbol{\varepsilon}^\star$ has positive entries). Thus isolating $\mathbf{P}^\star$ in the first order condition results in:

$$\mathbf{P}^\star = \mathrm{diag}(\mathbf{u}) \exp\left(-\mathrm{diag}(\boldsymbol{\varepsilon}^\star)^{-1}\mathbf{C}\right) \tag{13}$$

where $\mathbf{u} = \exp\left(\boldsymbol{\lambda}^\star \oslash \boldsymbol{\varepsilon}^\star\right)$. This matrix must satisfy the stochasticity constraint $\mathbf{P}\mathbf{1} = \mathbf{1}$. Hence one has $\mathbf{u} = \mathbf{1} \oslash \left(\exp\left(\mathrm{diag}(\boldsymbol{\varepsilon}^\star)^{-1}\mathbf{C}\right)\mathbf{1}\right)$ and $\mathbf{P}^\star$ has the form

$$\forall (i,j) \in [\![n]\!]^2, \quad P_{ij}^\star = \frac{\exp\left(-C_{ij}/\varepsilon_i^\star\right)}{\sum_\ell \exp\left(-C_{i\ell}/\varepsilon_i^\star\right)}. \tag{14}$$

As a consequence of $\boldsymbol{\varepsilon}^\star > \mathbf{0}$, complementary slackness in the KKT conditions gives us that for all $i$, the entropy constraint is saturated *i.e.*, $\mathrm{H}(\mathbf{P}_{i:}^\star) = \log \xi + 1$. Therefore $\mathbf{P}^\star$ solves the problem (EA). Conversely any solution of (EA) $P_{ij}^\star = \frac{\exp\left(-C_{ij}/\varepsilon_i^\star\right)}{\sum_\ell \exp\left(-C_{i\ell}/\varepsilon_i^\star\right)}$ with $(\varepsilon_i^\star)$ such that $\mathrm{H}(\mathbf{P}_{i:}^\star) = \log \xi + 1$ gives an admissible matrix for $\min_{\mathbf{P} \in \mathcal{H}_\xi} \langle \mathbf{P}, \mathbf{C} \rangle$ and the associated variables satisfy the KKT conditions which are sufficient conditions for optimality since the problem is convex. $\qquad\square$

## A.4  Proof of Proposition 4 and Proposition 5

The goal of this section is to prove the following results:

**Proposition 4** (Saturation of the entropies). *Let $\mathbf{C} \in \mathcal{S}$ with zero diagonal, then* (SEA) *with cost $\mathbf{C}$ has a* unique *solution that we denote by $\mathbf{P}^{\mathrm{se}}$. If moreover $\mathbf{C} \in \mathcal{D}$, then for at least $n-1$ indices $i \in [\![n]\!]$ the solution satisfies $\mathrm{H}(\mathbf{P}_{i:}^{\mathrm{se}}) = \log \xi + 1$.*

**Proposition 5** (Solving for SEA). *Let $\mathbf{C} \in \mathcal{D}, \mathcal{L}(\mathbf{P}, \boldsymbol{\gamma}, \boldsymbol{\lambda}) = \langle \mathbf{P}, \mathbf{C} \rangle + \langle \boldsymbol{\gamma}, (\log \xi + 1)\mathbf{1} - \mathrm{H_r}(\mathbf{P}) \rangle + \langle \boldsymbol{\lambda}, \mathbf{1} - \mathbf{P}\mathbf{1} \rangle$ and $q(\boldsymbol{\gamma}, \boldsymbol{\lambda}) = \min_{\mathbf{P} \in \mathbb{R}_+^{n \times n} \cap \mathcal{S}} \mathcal{L}(\mathbf{P}, \boldsymbol{\gamma}, \boldsymbol{\lambda})$. Strong duality holds for* (SEA). *Moreover, let $\boldsymbol{\gamma}^\star, \boldsymbol{\lambda}^\star \in \mathrm{argmax}_{\boldsymbol{\gamma} \geq 0, \boldsymbol{\lambda}} q(\boldsymbol{\gamma}, \boldsymbol{\lambda})$ be the optimal dual variables respectively associated with the entropy and marginal constraints. Then, for at least $n-1$ indices $i \in [\![n]\!], \gamma_i^\star > 0$. When $\forall i \in [\![n]\!], \gamma_i^\star > 0$ then $\mathrm{H_r}(\mathbf{P}^{\mathrm{se}}) = (\log \xi + 1)\mathbf{1}$ and $\mathbf{P}^{\mathrm{se}}$ has the form*

$$\mathbf{P}^{\mathrm{se}} = \exp\left((\boldsymbol{\lambda}^\star \oplus \boldsymbol{\lambda}^\star - 2\mathbf{C}) \oslash (\boldsymbol{\gamma}^\star \oplus \boldsymbol{\gamma}^\star)\right). \tag{2}$$

The unicity of the solution in Proposition 4 is a consequence of the following lemma

**Lemma 7.** *Let $\mathbf{C} \neq 0 \in \mathcal{S}$ with zero diagonal. Then the problem $\min_{\mathbf{P} \in \mathcal{H}_\xi \cap \mathcal{S}} \langle \mathbf{P}, \mathbf{C} \rangle$ has a unique solution.*

*Proof.* Making the constraints explicit, the primal problem of symmetric entropic affinity takes the following form

$$\min_{\mathbf{P} \in \mathbb{R}_+^{n \times n}} \quad \langle \mathbf{P}, \mathbf{C} \rangle$$
$$\text{s.t.} \quad \forall i, \ \mathrm{H}(\mathbf{P}_{i:}) \geq \log \xi + 1 \tag{SEA}$$
$$\mathbf{P}\mathbf{1} = \mathbf{1}, \quad \mathbf{P} = \mathbf{P}^\top.$$

Suppose that the solution is not unique *i.e.*, there exists a couple of optimal solutions $(\mathbf{P}_1, \mathbf{P}_2)$ that satisfy the constraints of (SEA) and such that $\langle \mathbf{P}_1, \mathbf{C} \rangle = \langle \mathbf{P}_2, \mathbf{C} \rangle$. For $i \in [\![n]\!]$, we denote the

function $f_i : \mathbf{P} \to (\log \xi + 1) - \mathrm{H}(\mathbf{P}_{i:})$. Then $f_i$ is continuous, strictly convex and the entropy conditions of (SEA) can be written as $\forall i \in [\![n]\!], f_i(\mathbf{P}) \leq 0$.

Now consider $\mathbf{Q} = \frac{1}{2}(\mathbf{P}_1 + \mathbf{P}_2)$. Then clearly $\mathbf{Q}\mathbf{1} = \mathbf{1}, \mathbf{Q} = \mathbf{Q}^\top$. Since $f_i$ is strictly convex we have $f_i(\mathbf{Q}) = f_i(\frac{1}{2}\mathbf{P}_1 + \frac{1}{2}\mathbf{P}_2) < \frac{1}{2}f_i(\mathbf{P}_1) + \frac{1}{2}f(\mathbf{P}_2) \leq 0$. Thus $f_i(\mathbf{Q}) < 0$ for any $i \in [\![n]\!]$. Take any $\varepsilon > 0$ and $i \in [\![n]\!]$. By continuity of $f_i$ there exists $\delta_i > 0$ such that, for any $\mathbf{H}$ with $\|\mathbf{H}\|_F \leq \delta_i$, we have $f_i(\mathbf{Q} + \mathbf{H}) < f_i(\mathbf{Q}) + \varepsilon$. Take $\varepsilon > 0$ such that $\forall i \in [\![n]\!], 0 < \varepsilon < -\frac{1}{2}f_i(\mathbf{Q})$ (this is possible since for any $i \in [\![n]\!], f_i(\mathbf{Q}) < 0$) and $\mathbf{H}$ with $\|\mathbf{H}\|_F \leq \min_{i \in [\![n]\!]} \delta_i$. Then for any $i \in [\![n]\!], f_i(\mathbf{Q} + \mathbf{H}) < 0$. In other words, we have proven that there exists $\eta > 0$ such that for any $\mathbf{H}$ such that $\|\mathbf{H}\|_F \leq \eta$, it holds: $\forall i \in [\![n]\!], f_i(\mathbf{Q} + \mathbf{H}) < 0$.

Now let us take $\mathbf{H}$ as the Laplacian matrix associated to $\mathbf{C}$ *i.e.*, for any $(i,j) \in [\![n]\!]^2$, $H_{ij} = -C_{ij}$ if $i \neq j$ and $\sum_l C_{il}$ otherwise. Then we have $\langle \mathbf{H}, \mathbf{C} \rangle = -\sum_{i \neq j} C_{ij}^2 + 0 = -\sum_{i \neq j} C_{ij}^2 < 0$ since $\mathbf{C}$ has zero diagonal (and is nonzero). Moreover, $\mathbf{H} = \mathbf{H}^\top$ since $\mathbf{C}$ is symmetric and $\mathbf{H}\mathbf{1} = \mathbf{0}$ by construction. Consider for $0 < \beta \leq \frac{\eta}{\|\mathbf{H}\|_F}$, the matrix $\mathbf{H}_\beta := \beta\mathbf{H}$. Then $\|\mathbf{H}_\beta\|_F = \beta\|\mathbf{H}\|_F \leq \eta$. By the previous reasoning one has: $\forall i \in [\![n]\!], f_i(\mathbf{Q} + \mathbf{H}_\beta) < 0$. Moreover, $(\mathbf{Q} + \mathbf{H}_\beta)^\top = \mathbf{Q} + \mathbf{H}_\beta$ and $(\mathbf{Q} + \mathbf{H}_\beta)\mathbf{1} = \mathbf{1}$. For $\beta$ small enough we have $\mathbf{Q} + \mathbf{H}_\beta \in \mathbb{R}_+^{n \times n}$ and thus there is a $\beta$ (that depends on $\mathbf{P}_1$ and $\mathbf{P}_2$) such that $\mathbf{Q} + \mathbf{H}_\beta$ is admissible *i.e.*, satisfies the constraints of (SEA). Then, for such $\beta$,

$$\langle \mathbf{C}, \mathbf{Q} + \mathbf{H}_\beta \rangle - \langle \mathbf{C}, \mathbf{P}_1 \rangle = \frac{1}{2}\langle \mathbf{C}, \mathbf{P}_1 + \mathbf{P}_2 \rangle + \langle \mathbf{C}, \mathbf{H}_\beta \rangle - \langle \mathbf{C}, \mathbf{P}_1 \rangle$$
$$= \langle \mathbf{C}, \mathbf{H}_\beta \rangle = \beta\langle \mathbf{H}, \mathbf{C} \rangle < 0 \,. \tag{15}$$

Thus $\langle \mathbf{C}, \mathbf{Q} + \mathbf{H}_\beta \rangle < \langle \mathbf{C}, \mathbf{P}_1 \rangle$ which leads to a contradiction. $\qquad\square$

We can now prove the rest of the claims of Proposition 4 and Proposition 5.

*Proof.* Let $\mathbf{C} \in \mathcal{D}$. We first prove Proposition 4. The unicity is a consequence of Lemma 7. For the saturation of the entropies we consider the Lagrangian of the problem (SEA) that writes

$$\mathcal{L}(\mathbf{P}, \boldsymbol{\lambda}, \boldsymbol{\gamma}, \boldsymbol{\Gamma}) = \langle \mathbf{P}, \mathbf{C} \rangle + \langle \boldsymbol{\gamma}, (\log \xi + 1)\mathbf{1} - \mathrm{H}_r(\mathbf{P}) \rangle + \langle \boldsymbol{\lambda}, \mathbf{1} - \mathbf{P}\mathbf{1} \rangle + \langle \boldsymbol{\Gamma}, \mathbf{P} - \mathbf{P}^\top \rangle$$

for dual variables $\boldsymbol{\gamma} \in \mathbb{R}_+^n$, $\boldsymbol{\lambda} \in \mathbb{R}^n$ and $\boldsymbol{\Gamma} \in \mathbb{R}^{n \times n}$. Strong duality holds by Slater's conditions because $\frac{1}{n}\mathbf{1}\mathbf{1}^\top$ is strictly feasible for $\xi \leq n - 1$. Since strong duality holds, for any solution $\mathbf{P}^\star$ to the primal problem and any solution $(\boldsymbol{\gamma}^\star, \boldsymbol{\lambda}^\star, \boldsymbol{\Gamma}^\star)$ to the dual problem, the pair $\mathbf{P}^\star, (\boldsymbol{\gamma}^\star, \boldsymbol{\lambda}^\star, \boldsymbol{\Gamma}^\star)$ must satisfy the KKT conditions. They can be stated as follows:

$$\mathbf{C} + \mathrm{diag}(\boldsymbol{\gamma}^\star) \log \mathbf{P}^\star - \boldsymbol{\lambda}^\star \mathbf{1}^\top + \boldsymbol{\Gamma}^\star - \boldsymbol{\Gamma}^{\star\top} = \mathbf{0}$$
$$\mathbf{P}^\star\mathbf{1} = \mathbf{1}, \mathrm{H}_r(\mathbf{P}^\star) \geq (\log \xi + 1)\mathbf{1}, \mathbf{P}^\star = \mathbf{P}^{\star\top}$$
$$\boldsymbol{\gamma}^\star \geq \mathbf{0} \tag{KKT-SEA}$$
$$\forall i, \gamma_i^\star(\mathrm{H}(\mathbf{P}_{i:}^\star) - (\log \xi + 1)) = 0 \,.$$

Let us denote $I = \{\ell \in [\![n]\!] \text{ s.t. } \gamma_\ell^\star = 0\}$. For $\ell \in I$, using the first-order condition, one has for $i \in [\![n]\!], C_{\ell i} = \lambda_\ell^\star - \Gamma_{\ell i}^\star + \Gamma_{i\ell}^\star$. Since $\mathbf{C} \in \mathcal{D}$, we have $C_{\ell\ell} = 0$ thus $\lambda_\ell^\star = 0$ and $C_{\ell i} = \Gamma_{i\ell}^\star - \Gamma_{\ell i}^\star$. For $(\ell, \ell') \in I^2$, one has $C_{\ell\ell'} = \Gamma_{\ell'\ell}^\star - \Gamma_{\ell\ell'}^\star = -(\Gamma_{\ell\ell'}^\star - \Gamma_{\ell'\ell}^\star) = -C_{\ell'\ell}$. $\mathbf{C}$ is symmetric thus $C_{\ell\ell'} = 0$. Since $\mathbf{C}$ only has null entries on the diagonal, this shows that $\ell = \ell'$ and therefore $I$ has at most one element. By complementary slackness condition (last row of the KKT-SEA conditions) it holds that $\forall i \neq \ell, \mathrm{H}(\mathbf{P}_{i:}^\star) = \log \xi + 1$. Since the solution of (SEA) is unique $\mathbf{P}^\star = \mathbf{P}^{\mathrm{se}}$ and thus $\forall i \neq \ell, \mathrm{H}(\mathbf{P}_{i:}^{\mathrm{se}}) = \log \xi + 1$ which proves Proposition 4 but also that for at least $n - 1$ indices $\gamma_i^\star > 0$. Moreover, from the KKT conditions we have

$$\forall (i,j) \in [\![n]\!]^2, \ \Gamma_{ji}^\star - \Gamma_{ij}^\star = C_{ij} + \gamma_i^\star \log P_{ij}^\star - \lambda_i^\star \,. \tag{16}$$

Now take $(i,j) \in [\![n]\!]^2$ fixed. From the previous equality $\Gamma_{ji}^\star - \Gamma_{ij}^\star = C_{ij} + \gamma_i^\star \log P_{ij}^\star - \lambda_i^\star$ but also $\Gamma_{ij}^\star - \Gamma_{ji}^\star = C_{ji} + \gamma_j^\star \log P_{ji}^\star - \lambda_j^\star$. Using that $\mathbf{P}^\star = (\mathbf{P}^\star)^\top$ and $\mathbf{C} \in \mathcal{S}$ we get $\Gamma_{ij}^\star - \Gamma_{ji}^\star = C_{ij} + \gamma_j^\star \log P_{ij}^\star - \lambda_j^\star$. But $\Gamma_{ij}^\star - \Gamma_{ji}^\star = -(\Gamma_{ji}^\star - \Gamma_{ij}^\star)$ which gives

$$C_{ij} + \gamma_j^\star \log P_{ij}^\star - \lambda_j^\star = -(C_{ij} + \gamma_i^\star \log P_{ij}^\star - \lambda_i^\star) \,. \tag{17}$$

This implies

$$\forall (i,j) \in [\![n]\!]^2, \ 2C_{ij} + (\gamma_i^\star + \gamma_j^\star) \log P_{ij}^\star - (\lambda_i^\star + \lambda_j^\star) = 0 \,. \tag{18}$$

Consequently, if $\boldsymbol{\gamma}^\star > 0$ we have the desired form from the above equation and by complementary slackness $\mathrm{H_r}(\mathbf{P}^{\mathrm{se}}) = (\log \xi + 1)\mathbf{1}$ which proves Proposition 5. Note that otherwise, it holds

$$\forall (i,j) \neq (\ell, \ell), \ P_{ij}^\star = \exp \left( \frac{\lambda_i^\star + \lambda_j^\star - 2C_{ij}}{\gamma_i^\star + \gamma_j^\star} \right) \,. \tag{19}$$

$\square$

## A.5 EA and SEA as a KL projection

We prove the characterization as a projection of (EA) in Lemma 8 and of (SEA) in Lemma 9.

**Lemma 8.** *Let* $\mathbf{C} \in \mathcal{D}, \sigma > 0$ *and* $\mathbf{K}_\sigma = \exp(-\mathbf{C}/\sigma)$. *Then for any* $\sigma \leq \min_i \varepsilon_i^\star$, *it holds* $\mathbf{P}^{\mathrm{e}} = \mathrm{Proj}_{\mathcal{H}_\xi}^{\mathrm{KL}}(\mathbf{K}_\sigma) = \arg\min_{\mathbf{P} \in \mathcal{H}_\xi} \mathrm{KL}(\mathbf{P}|\mathbf{K}_\sigma)$.

*Proof.* The KL projection of $\mathbf{K}$ onto $\mathcal{H}_\xi$ reads

$$\min_{\mathbf{P} \in \mathbb{R}_+^{n \times n}} \quad \mathrm{KL}(\mathbf{P}|\mathbf{K}) \tag{20}$$

$$\text{s.t.} \quad \forall i, \ \mathrm{H}(\mathbf{P}_{i:}) \geq \log \xi + 1 \tag{21}$$

$$\mathbf{P1} = \mathbf{1} \,. \tag{22}$$

Introducing the dual variables $\boldsymbol{\lambda} \in \mathbb{R}^n$ and $\boldsymbol{\kappa} \in \mathbb{R}_+^n$, the Lagrangian of this problem reads:

$$\mathcal{L}(\mathbf{P}, \boldsymbol{\lambda}, \boldsymbol{\kappa}) = \mathrm{KL}(\mathbf{P}|\mathbf{K}) + \langle \boldsymbol{\kappa}, (\log \xi + 1)\mathbf{1} - \mathrm{H}(\mathbf{P}) \rangle + \langle \boldsymbol{\lambda}, \mathbf{1} - \mathbf{P1} \rangle \tag{23}$$

Strong duality holds hence for any solution $\mathbf{P}^\star$ to the above primal problem and any solution $(\boldsymbol{\kappa}^\star, \boldsymbol{\lambda}^\star)$ to the dual problem, the pair $\mathbf{P}^\star, (\boldsymbol{\kappa}^\star, \boldsymbol{\lambda}^\star)$ must satisfy the KKT conditions. The first-order optimality condition gives:

$$\nabla_{\mathbf{P}} \mathcal{L}(\mathbf{P}^\star, \boldsymbol{\kappa}^\star, \boldsymbol{\lambda}^\star) = \log(\mathbf{P}^\star \oslash \mathbf{K}) + \mathrm{diag}(\boldsymbol{\kappa}^\star) \log \mathbf{P}^\star - \boldsymbol{\lambda}^\star \mathbf{1}^\top = \mathbf{0} \,. \tag{24}$$

Solving for $\boldsymbol{\lambda}^\star$ given the stochasticity constraint and isolating $\mathbf{P}^\star$ gives

$$\forall (i,j) \in [\![n]\!]^2, \quad P_{ij}^\star = \frac{\exp\left((\log K_{ij})/(1 + \kappa_i^\star)\right)}{\sum_\ell \exp\left((\log K_{i\ell})/(1 + \kappa_i^\star)\right)} \,. \tag{25}$$

We now consider $\mathbf{P}^\star$ as a function of $\boldsymbol{\kappa}$. Plugging this expression back in $\mathcal{L}$ yields the dual function $\boldsymbol{\kappa} \mapsto \mathcal{G}(\boldsymbol{\kappa})$. The latter is concave as any dual function and its gradient reads:

$$\nabla_{\boldsymbol{\kappa}} \mathcal{G}(\boldsymbol{\kappa}) = (\log \xi + 1)\mathbf{1} - \mathrm{H}(\mathbf{P}^\star(\boldsymbol{\kappa})) \,. \tag{26}$$

Denoting by $\boldsymbol{\rho} = \mathbf{1} + \boldsymbol{\kappa}$ and taking the dual feasibility constraint $\boldsymbol{\kappa} \geq \mathbf{0}$ into account gives the solution: for any $i$, $\rho_i^\star = \max(\varepsilon_i^\star, 1)$ where $\boldsymbol{\varepsilon}^\star$ solves (EA) with cost $\mathbf{C} = -\log \mathbf{K}$. Moreover we have that $\sigma \leq \min(\boldsymbol{\varepsilon}^\star)$ where $\boldsymbol{\varepsilon}^\star \in (\mathbb{R}_+^*)^n$ solves (EA). Therefore for any $i \in [\![n]\!]$, one has $\varepsilon_i^\star/\sigma \geq 1$. Thus there exists $\kappa_i^\star \in \mathbb{R}_+$ such that $\sigma(1 + \kappa_i^\star) = \varepsilon_i^\star$.

This $\boldsymbol{\kappa}^\star$ cancels the above gradient *i.e.*, $(\log \xi + 1)\mathbf{1} = \mathrm{H}(\mathbf{P}^\star(\boldsymbol{\kappa}^\star))$ thus solves the dual problem. Therefore given the expression of $\mathbf{P}^\star$ we have that $\mathrm{Proj}_{\mathcal{H}_\xi}^{\mathrm{KL}}(\mathbf{K}) = \mathbf{P}^{\mathrm{e}}$. $\square$

**Lemma 9.** *Let* $\mathbf{C} \in \mathcal{D}, \sigma > 0$ *and* $\mathbf{K}_\sigma = \exp(-\mathbf{C}/\sigma)$. *Suppose that the optimal dual variable* $\gamma^\star$ *associated with the entropy constraint of* (SEA) *is positive. Then for any* $\sigma \leq \min_i \gamma_i^\star$, *it holds* $\mathbf{P}^{\mathrm{se}} = \mathrm{Proj}_{\mathcal{H}_\xi \cap \mathcal{S}}^{\mathrm{KL}}(\mathbf{K}_\sigma)$.

*Proof.* Let $\sigma > 0$. The KL projection of $\mathbf{K}$ onto $\mathcal{H}_\xi \cap \mathcal{S}$ boils down to the following optimization problem.

$$\min_{\mathbf{P} \in \mathbb{R}_+^{n \times n}} \quad \mathrm{KL}(\mathbf{P}|\mathbf{K}_\sigma)$$

$$\text{s.t.} \quad \forall i, \ \mathrm{H}(\mathbf{P}_{i:}) \geq \log \xi + 1 \tag{SEA-Proj}$$

$$\mathbf{P1} = \mathbf{1}, \quad \mathbf{P}^\top = \mathbf{P} \,.$$

By strong convexity of $\mathbf{P} \to \mathrm{KL}(\mathbf{P}|\mathbf{K}_\sigma)$ and convexity of the constraints the problem (SEA-Proj) admits a unique solution. Moreover, the Lagrangian of this problem takes the following form, where $\boldsymbol{\omega} \in \mathbb{R}_+^n, \boldsymbol{\mu} \in \mathbb{R}^n$ and $\boldsymbol{\Gamma} \in \mathbb{R}^{n \times n}$:

$$\mathcal{L}(\mathbf{P}, \boldsymbol{\mu}, \boldsymbol{\omega}, \boldsymbol{\Gamma}) = \mathrm{KL}(\mathbf{P}|\mathbf{K}_\sigma) + \langle \boldsymbol{\omega}, (\log \xi + 1)\mathbf{1} - \mathrm{H}_r(\mathbf{P}) \rangle + \langle \boldsymbol{\mu}, \mathbf{1} - \mathbf{P}\mathbf{1} \rangle + \langle \boldsymbol{\beta}, \mathbf{P} - \mathbf{P}^\top \rangle \,.$$

Strong duality holds by Slater's conditions thus the KKT conditions are necessary and sufficient. In particular if $\mathbf{P}^\star$ and $(\boldsymbol{\omega}^\star, \boldsymbol{\mu}^\star, \boldsymbol{\beta}^\star)$ satisfy

$$\nabla_{\mathbf{P}}\mathcal{L}(\mathbf{P}^\star, \boldsymbol{\mu}^\star, \boldsymbol{\omega}^\star, \boldsymbol{\Gamma}^\star) = \log(\mathbf{P}^\star \oslash \mathbf{K}) + \mathrm{diag}(\boldsymbol{\omega}^\star)\log \mathbf{P}^\star - \boldsymbol{\mu}^\star\mathbf{1}^\top + \boldsymbol{\beta}^\star - \boldsymbol{\beta}^{\star\top} = \mathbf{0}$$

$$\mathbf{P}^\star\mathbf{1} = \mathbf{1}, \; \mathrm{H}_r(\mathbf{P}^\star) \geq (\log\xi + 1)\mathbf{1}, \; \mathbf{P}^\star = \mathbf{P}^{\star\top} \qquad \text{(KKT-Proj)}$$

$$\boldsymbol{\omega}^\star \geq \mathbf{0}$$

$$\forall i, \omega_i^\star(\mathrm{H}(\mathbf{P}_{i:}^\star) - (\log\xi + 1)) = 0 \,.$$

then $\mathbf{P}^\star$ is a solution to (SEA-Proj) and $(\boldsymbol{\omega}^\star, \boldsymbol{\mu}^\star, \boldsymbol{\beta}^\star)$ are optimal dual variables. The first condition rewrites

$$\forall(i,j), \; \log(P_{ij}^\star) + \frac{1}{\sigma}C_{ij} + \omega_i^\star \log(P_{ij}^\star) - \mu_i^\star + \beta_{ij}^\star - \beta_{ji}^\star = 0 \,, \qquad (27)$$

which is equivalent to

$$\forall(i,j), \; \sigma(1 + \omega_i^\star)\log(P_{ij}^\star) + C_{ij} - \sigma\mu_i^\star + \sigma(\beta_{ij}^\star - \beta_{ji}^\star) = 0 \,. \qquad (28)$$

Now take $\mathbf{P}^{\mathrm{se}}$ the optimal solution of (SEA). As written in the proof Proposition 5 of $\mathbf{P}^{\mathrm{se}}$ and the optimal dual variables $(\boldsymbol{\gamma}^\star, \boldsymbol{\lambda}^\star, \boldsymbol{\Gamma}^\star)$ satisfy the KKT conditions:

$$\forall(i,j), \; C_{ij} + \gamma_i^\star \log P_{ij}^{\mathrm{se}} - \lambda_i^\star + \Gamma_{ij}^\star - \Gamma_{ji}^\star = \mathbf{0}$$

$$\mathbf{P}^{\mathrm{se}}\mathbf{1} = \mathbf{1}, \; \mathrm{H}_r(\mathbf{P}^{\mathrm{se}}) \geq (\log\xi + 1)\mathbf{1}, \; \mathbf{P}^{\mathrm{se}} = (\mathbf{P}^{\mathrm{se}})^\top \qquad \text{(KKT-SEA)}$$

$$\boldsymbol{\gamma}^\star \geq \mathbf{0}$$

$$\forall i, \gamma_i^\star(\mathrm{H}(\mathbf{P}_{i:}^{\mathrm{se}}) - (\log\xi + 1)) = 0 \,.$$

By hypothesis $\boldsymbol{\gamma}^\star > 0$ which gives $\forall i, \mathrm{H}(\mathbf{P}_{i:}^{\mathrm{se}}) - (\log\xi + 1) = 0$. Now take $0 < \sigma \leq \min_i \gamma_i^\star$ and define $\forall i, \omega_i^\star = \frac{\gamma_i^\star}{\sigma} - 1$. Using the hypothesis on $\sigma$ we have $\forall i, \omega_i^\star \geq 0$ and $\boldsymbol{\omega}^\star$ satisfies $\forall i, \; \sigma(1 + \omega_i^\star) = \gamma_i^\star$. Moreover for any $i \in [\![n]\!]$

$$\omega_i^\star(\mathrm{H}(\mathbf{P}_{i:}^{\mathrm{se}}) - (\log\xi + 1)) = 0 \,. \qquad (29)$$

Define also $\forall i, \mu_i^\star = \lambda_i^\star/\sigma$ and $\forall(i,j), \beta_{ij}^\star = \Gamma_{ij}^\star/\sigma$. Since $\mathbf{P}^{\mathrm{se}}, (\boldsymbol{\gamma}^\star, \boldsymbol{\lambda}^\star, \boldsymbol{\Gamma}^\star)$ satisfies the KKT conditions (KKT-SEA) then by the previous reasoning $\mathbf{P}^{\mathrm{se}}, (\boldsymbol{\omega}^\star, \boldsymbol{\mu}^\star, \boldsymbol{\beta}^\star)$ satisfy the KKT conditions (KKT-Proj) and in particular $\mathbf{P}^{\mathrm{se}}$ is an optimal solution of (SEA-Proj) since KKT conditions are sufficient. Thus we have proven that $\mathbf{P}^{\mathrm{se}} \in \arg\min_{\mathbf{P} \in \mathcal{H}_\xi \cap \mathcal{S}} \mathrm{KL}(\mathbf{P}|\mathbf{K}_\sigma)$ and by the uniqueness of the solution this is in fact an equality. $\qquad \square$

## B   Sensitivity Analysis for Dimensionality Reduction Experiments

In Figure 8, we extend the sensitivity analysis performed for spectral clustering (Figure 5) to DR scores. One can notice that tSNEkhorn outperforms tSNE on a wide range of perplexity values.

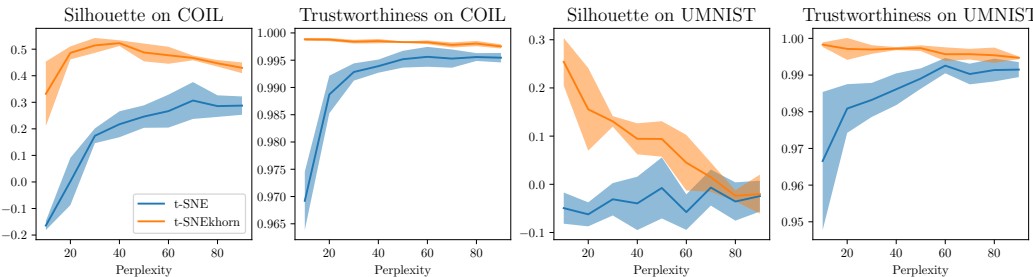

Figure 8: Dimensionality reduction scores as a function of the perplexity parameter.

