# OpenReview forum: "SNEkhorn: Dimension Reduction with Symmetric Entropic Affinities"
_NeurIPS.cc/2023/Conference — NeurIPS 2023 poster_

### Official Review · Reviewer_8Btp · 2023-07-06

**Soundness:** 4 excellent
**Presentation:** 3 good
**Contribution:** 3 good
**Rating:** 6
**Confidence:** 3

**Summary:**

This paper presents a new dimensionality reduction algorithm, named SNEkhorn. By uncovering the novel links between Entropic affinities (EAs) and Optimal Transport (OT), the authors derive EAs with symmetric doubly stochastic normalization and the fixed row-wise entropy, which is the key to SNEkhorn. Besides, the authors show a dual ascent algorithm to compute this new affinity efficiently. Beyond this theoretical contribution, the authors showcase the benefits of SNEkhorn through numerical experiments on simulated data and real data.

**Strengths:**

- Entropic affinities (EAs) needs to be symmetrized when used in popular Dimensionality Reduction (DR) algorithm t-SNE, but the symmetrization destroys the row-wise constant entropy and stochasticity properties of EAs. To derive EAs that can maintain both symmetry and its own properties is a very natural. The proposed method is novel and simple to describe.
- The authors discover novel connections between EAs and OT, and show a dual ascent algorithm to compute EAs from OT. The proposed algorithm provides a new perspective on EAs.
- For conventional DR algorithm, the latent affinity is not doubly stochastic, which imposes spherical constraints on the embedding space. The proposed methods avoid this problem.
- The simulation study and real data analysis are very sufficient. The authors illustrate the effectiveness from many aspects.

**Weaknesses:**

- There is not any discussion about the computational complexity of proposed method. Although the proposed algorithm performs better than t-SNE and UMAP, how does the computational complexity of proposed method compare with t-SNE or UMAP. Since the computational cost is also important in practice.
- Some of the formulas and references in the article are incorrectly hyperlinked.

**Questions:**

- How does the computational complexity for the proposed method compare to t-SNE? Does the proposed method spend the same amount of time as t-SNE while achieve better results?
- How to choose the dimensionality of PCA? In the reference the authors mentioned, the dimensionality of PCA is usually set to be 30 instead of 50. Does the dimensionality affects the performance of the algorithm?

**Limitations:**

Yes

---

> ### Author Rebuttal · Authors · 2023-08-08
>
> We thank the reviewer for the careful reading of the manuscript, her/his assessment and relevant remarks.
>
> > How does the computational complexity for the proposed method compare to t-SNE? Does the proposed method spend the same amount of time as t-SNE while achieve better results?
>
> This is a very good point. We provide some additional insights about computational aspects in the general answers as well as some runtime experiments in the pdf. Although both methods have a complexity that scales quadratically with the number of samples, one can notice in these experiments that the current implementation of t-SNEkhorn is slower than t-SNE. We will add a discussion and computational times in the final version of the paper.
>
> Importantly, note that in our paper we did not use the most recent acceleration procedures that have been proposed for t-SNE [2,3] but our approach will directly benefit from many of those computational tricks.
>
> > How to choose the dimensionality of PCA? In the reference the authors mentioned, the dimensionality of PCA is usually set to be 30 instead of 50. Does the dimensionality affects the performance of the algorithm?
>
> This point indeed deserves interest as PCA is often applied as a pre-processing step before computing the distance matrix. Choosing the dimensionality of this PCA step seems to depend on the nature of the data (see e.g. [4] for scRNA data that recommends using 50 axes). In his t-SNE website (unlike in the paper as the reviewer rightfully pointed out), Laurens Van der Maaten also recommends using 50 axes. In our experiments, as shown for instance in table 2 of the pdf above provided with the global response, we did not find any significant difference between using 30 axes and 50 axes (although this might not be true for all datasets).
>
> [2] Laurens Van Der Maaten. Barnes-hut-sne. arXiv preprint arXiv:1301.3342, 2013.
>
> [3] Linderman, G. C., Rachh, M., Hoskins, J. G., Steinerberger, S. and Kluger, Y. (2019). Fast interpolation-based t-SNE for improved vi- sualization of single-cell RNA-seq data. Nature Methods 16 243-245.
>
> [4] Dmitry Kobak and Philipp Berens. The art of using t-SNE for single-cell transcriptomics.
> Nature Communication, 10:5416, 2019.

---

### Official Review · Reviewer_W2ip · 2023-07-07

**Soundness:** 4 excellent
**Presentation:** 3 good
**Contribution:** 4 excellent
**Rating:** 8
**Confidence:** 5

**Summary:**

This is a very interesting paper about an application of the Sinkhorn algorithm to symmetrize the matrix of entropic affinities in methods of DR like SNE, t-SNE, etc.
Strong theoretical contribution.
Some experiments to illustrate.

**Strengths:**

The paper is very interesting for its vision, state of the art across multiple domains, strong theoretical contribution, revisiting of existing methods and proposal of a new method.
The paper is very well written, not rushed.
The symmetrization with Sinkhorn is well done and this a nice "tour de force" like only Brittons can do.
The experiments include a grid search on the perplexity, and multiple runs for each perplexity.

**Weaknesses:**

The paper is well written but it is also very/too dense: much information is packed, that is nice, but it takes quite some time to digest and to decode.
The notation is sometimes unconventional and difficult to follow, probably to keep it as compact as possible.
While the theoretical part is very convincing and interesting for itself, it will only appeal to a very restricted audience of people specialised in the methodological design of advanced methods of DR. Also, the experiments are not sufficient to convince practitioners that the Sinkhorn symmetrization is from now on something necessary to get the best results. The experiments lack broader comparisons with other methods that are in the state of the art (here it looks like the authors have reimplemented t-SNE and UMAP; also, there are methods that do not look for symmetry at all, although this can be detrimental in the presence of many isolated outliers). The quantitative assessment might be partly questionable: the trustworthiness is not the best DR QA indicator; what was the neighbourhood size, by the way?
The lack of accelerated version is a possible weakness for dissemination, while NeurIPS would be precisely the ideal launching pad.

**Questions:**

The "crowding effect" is just an intuitive explanation found in LvdM t-SNE paper; would it not be better to refer to something more conventional (empty space phenomenon, norm concentration, or just the reduction in volume from HD to LD?)?
In Figure 1, SNEkhorn or t-SNEkhorn ?

**Limitations:**

No possible negative societal impact here.

---

> ### Author Rebuttal · Authors · 2023-08-08
>
> We would like to thank the reviewer for his/her careful reading of the manuscript and insightful comments.
>
> > The quantitative assessment might be partly questionable: the trustworthiness is not the best DR QA indicator; what was the neighbourhood size, by the way?
>
> This is an interesting point. We wanted to fairly evaluate performances of the DR methods methods and to avoid ad-hoc performance measures that are used in the optimization problem of specific methods. This is why we chose classical (if a bit unorthodox) performance measures that evaluate the preservation of global (silhouette score) and local (trustworthiness) structures from the data. We are interested if the reviewer suggests an alternative that we might add in supplementary. For trustworthiness, the neighbourhood size was set to the default value of 5.
>
> > The "crowding effect" is just an intuitive explanation found in LvdM t-SNE paper; would it not be better to refer to something more conventional (empty space phenomenon, norm concentration, or just the reduction in volume from HD to LD?)? In Figure 1, SNEkhorn or t-SNEkhorn ?
>
> The reviewer is raising a good point. We mentionned the "crowding effect" to be faithful to the story of t-SNE and explain how heavy-tailed kernel first appeared in these methods. We agree indeed that a more conventional explanation will be better and will modify our article accordingly for the sake of clarity.
>
> For figure 1, we indeed used SNEkhorn in the embedding space as we found that it gave descent results on the COIL dataset and we lacked space to provide both SNEkhorn and t-SNEkhorn embeddings. We will plot both methods in the supplementary in the final version of the paper.
>
> Note that quantitative results for t-SNE and t-SNEkhorn on COIL are provided in table 3.

---

### Official Review · Reviewer_s51e · 2023-07-27

**Soundness:** 4 excellent
**Presentation:** 3 good
**Contribution:** 3 good
**Rating:** 7
**Confidence:** 3

**Summary:**

The paper presents a novel approach to dealing with entropic affinities (EAs) used in machine learning for dimensionality reduction tasks, specifically in the popular t-SNE algorithm. It addresses the limitations of current symmetrization methods applied to EAs, which can compromise the entropy and stochasticity properties of the affinity matrix. The proposed method uses optimal transport to achieve a natural symmetrization, leading to a new affinity matrix. This new matrix is then leveraged in a new dimensionality reduction algorithm called SNEkhorn, which is demonstrated to outperform state-of-the-art methods on both synthetic and real-world datasets.

**Strengths:**

1. The paper introduces a novel approach to symmetrizing entropic affinities by formulating them as an optimal transport problem. This brings a fresh perspective to the problem and could open up new avenues for research and application in machine learning.
2. The proposed symmetrization method claims to maintain the constant entropy and stochasticity properties of the affinity matrix while being computationally efficient through dual ascent. This combination of robustness and efficiency is essential for practical applications.
3. The development of the SNEkhorn algorithm based on the new affinity matrix offers a practical implementation of the proposed approach. The demonstrated superiority of SNEkhorn over other baseline methods on various datasets further strengthens the paper's claims.
4. The paper evaluates the proposed SNEkhorn algorithm on both synthetic and real-world datasets, providing a comprehensive assessment of its performance and general applicability.

**Weaknesses:**

Clarity of Presentation: While the abstract provides a high-level overview, some concepts, such as entropic affinities, optimal transport, and dual ascent, might be challenging for readers not already familiar with the domain. The paper should provide a clear and concise introduction to these concepts, ideally with intuitive explanations and illustrative examples. For example, what is P and C on paper 3 "Symmetric Entropy-Constrained Optimal Transport"

Robustness Analysis: Since the proposed method claims to be robust to varying noise levels, it would be valuable to include a thorough analysis of its performance under different levels of noise in the datasets instead of two level. This could strengthen the claim of robustness and highlight the algorithm's practicality in real-world scenarios.


**Questions:**

1. What sensitivity analysis was performed on the hyperparameters introduced by the new symmetrization approach? How robust is the SNEkhorn algorithm to changes in these parameters?

2. Can you provide additional insights into the interpretability of the reduced representations obtained by SNEkhorn? How well does it preserve meaningful structure in the data compared to other methods? Is the visualization in Figure 6 a special case?

3.Could you elaborate on the limitations or potential failure cases of the proposed approach? What are the scenarios where SNEkhorn might not perform as well or could face challenges?

**Limitations:**

While the paper proposes a novel approach to symmetrizing entropic affinities and introduces the SNEkhorn algorithm with promising results, it is essential to consider the potential limitations and shortcomings of the work. Some possible limitations include:

Scalability: Dimensionality reduction algorithms often face challenges with scalability when dealing with large datasets. The paper should address the computational efficiency and scalability of the proposed SNEkhorn algorithm, especially when applied to high-dimensional and massive datasets commonly encountered in real-world applications.

Parameter Sensitivity: It is essential to investigate the sensitivity of hyper parameters and assess how they impact the performance of the SNEkhorn algorithm, e.g. perplexity.

---

> ### Author Rebuttal · Authors · 2023-08-08
>
> We would like to thank the reviewer for the appreciation of our work, insightful comments and questions.
>
> ### Answers to weaknesses:
>
> > Clarity of Presentation: While the abstract provides a high-level overview, some concepts, such as entropic affinities, optimal transport, and dual ascent, might be challenging for readers not already familiar with the domain. The paper should provide a clear and concise introduction to these concepts, ideally with intuitive explanations and illustrative examples. For example, what is P and C on paper 3 "Symmetric Entropy-Constrained Optimal Transport"
>
> We will add details about these concepts in the paper as it may indeed appear challenging for some readers. Note that in our article C is the transportation cost matrix (C_{ij} is the cost of transporting a unit of mass from sample i to sample j) while P is the transport plan (P_{ij} is the amount of mass transported from sample i to sample j) that we propose to use as affinity matrix. We will make it clearer in the revised version.
>
> > Robustness Analysis: Since the proposed method claims to be robust to varying noise levels, it would be valuable to include a thorough analysis of its performance under different levels of noise in the datasets instead of two level. This could strengthen the claim of robustness and highlight the algorithm's practicality in real-world scenarios.
>
> This is a very interesting question. We provide some new results studying the robustness to noise in the global reply (Figure 1 in the pdf).
>
> ### Answers to questions
>
> > What sensitivity analysis was performed on the hyperparameters introduced by the new symmetrization approach? How robust is the SNEkhorn algorithm to changes in these parameters?
>
> We provide a sensitivity analysis in Fig. 5 in the paper for spectral clustering applications. We also added in the global reply (see Figure 2 in the adjoining PDF) a new sensitivity analysis for the end task of dimensionality reduction. This shows the superiority of t-SNEkhorn over t-SNE over a wide range of perplexity values.
>
> > Can you provide additional insights into the interpretability of the reduced representations obtained by SNEkhorn? How well does it preserve meaningful structure in the data compared to other methods? Is the visualization in Figure 6 a special case?
>
> Similarly to SNE (or UMAP), SNEkhorn's axis cannot be directly interpreted due to the non-linearity of the model.
> As shown in section 5, using symmetric entropic affinities and SNEkhorn lead to better robustness to heteroscedastic noise as well as enhanced clustering abilities compared to previous approaches. As such, the partitioning structure as well as close range pairwise relations are better preserved in SNEkhorn.
> We displayed figure 1 and 6 to give a visual interpretation of the figures given in table 3. It is indeed an example where SNEkhorn performs much better than other methods.
>
> > 3.Could you elaborate on the limitations or potential failure cases of the proposed approach? What are the scenarios where SNEkhorn might not perform as well or could face challenges?
>
> First note that while our proposed symmetric EAs are much more robust than the l2 symmetrization carried out in t-SNE, we will share many failure cases with tSNE, i.e. when the perplexity is too large (loss of local subtleties) or too small (fake structures appearing). In addition, our method comes with a more involved computational cost since we need to optimize twice the number of parameters (vectors $\boldsymbol\gamma$ and $\boldsymbol\lambda$) for affinity matrix estimation (instead of only $\boldsymbol\epsilon$ for the directed EAs). For runtime values, we refer to table 1 of the pdf in the global answer above.

---

### Official Review · Reviewer_WhX7 · 2023-08-03

**Soundness:** 3 good
**Presentation:** 3 good
**Contribution:** 3 good
**Rating:** 6
**Confidence:** 4

**Summary:**

Existing DR approaches employ EAs after heuristic symmetrisation (symmetric-SNE). This leads to less faithful embeddings with low silhouette scores. This paper avoids such heuristic symmetrisation by enforcing symmetrisation in a related, new OT based formulation.

Towards this goal, firstly, the EA problem is equivalently written as a (semi-relaxed) OT problem, where the transport plan recovers the EAs. Then symmetrisation is explicitly enforced as constraints on the transport map in this OT problem. Using optimality conditions it is shown that such EAs, maintain the crucial row-wise entropic equalities (prop4). As a result EAs are symmetric, doubly stochastic, and satisfy the entropic equalities, making the organically superior to the existing heuristics.

Using the proposed EAs, a DR formulation is proposed (leading to so-called SNEkhorn). Details of solving this problem are presented. The proposed embeddings are empirically compared to the state-of-the-art wrt. spectral clustering and dimension reduction. The improvements are significant.

**Strengths:**

1. The organisation and write-up are well polished and makes it an easy read.
2. Connections made between EAs and OT are interesting, especially in the light of the symmetrisation issue with existing DR approach.
3. Empirical improvements over baselines are impressive.


**Weaknesses:**

..

**Questions:**

1. It seems a primary reason for having Q_Z^{ds} in SNEkhorn is to enable fast objective-gradient computation via sinkhorn. Is this true?
2. Will it be more appropriate to consider other affinities like \tilde{Q}_Z etc. in place of Q_Z^{ds} ? If \tilde{Q}_Z leads to spherical embeddings, can there be applications where this is desirable? Any discussion on this and alternatives may help the reader.



**Limitations:**

Limitations were discussed in the concluding section.

---

> ### Author Rebuttal · Authors · 2023-08-08
>
> We would like to thank the reviewer for her/his insightful comments and questions. Our answers to the questions raised can be found below.
>
> > It seems a primary reason for having Q_Z^{ds} in SNEkhorn is to enable fast objective-gradient computation via sinkhorn. Is this true?
>
> It is true that it enables fast objective-gradient computation via the Sinkhorn algorithm. However the primary reason for using such Q_Z^{ds} is to counter the sphere concentration phenomenon for the embedding occuring when matching a non DS affinity (for instance \tilde{Q}_Z or Q_Z) with a DS affinity for input data [1]. This geometrical effect disapears when the embeddings affinity is itself DS. This is very useful as soon as one wants to embed onto a flat space.
>
> > Will it be more appropriate to consider other affinities like \tilde{Q}_Z etc. in place of Q_Z^{ds} ? If \tilde{Q}_Z leads to spherical embeddings, can there be applications where this is desirable? Any discussion on this and alternatives may help the reader.
>
> Thank you for this very interesting comment. Spherical embeddings are desirable if the embedding space is indeed a sphere. This is what is done in [1]. In our work, we rather focused on Euclidean flat spaces as it is what practionners are usually interested in when performing dimensionality reduction (for instance with scRNA seq data). Note that this also suggests as future works to study SNEkhorn for embedding on spherical manifolds. This discussion will be added in the paper.
>
> [1] Yao Lu, Jukka Corander, and Zhirong Yang. Doubly stochastic neighbor embedding on spheres. Pattern Recognition Letters, 128:100–106, 2019.

---

> ### Comment · Reviewer_WhX7 · 2023-08-18
>
> Thanks for the reply. I would like to keep my score after reading the rebuttal and going through other reviews.

---

### Author Rebuttal · Authors · 2023-08-08

We first would like to thank all the reviewers for their remarks and questions.

You may find attached a pdf with some new results to answer the various points raised. The new results are as follows.

## New results

### Figure 1 : Robustness to noise

In Figure 1, we focus on reviewer s51e's question about the robustness analysis. We plot spectral clustering scores on the clustering of three Gaussian clusters with variance $\sigma$, $2\sigma$ and $3\sigma$. This figure shows that symmetric entropic affinities are more robust than the Sinkhorn kernel when $\sigma$ increases. Note that the later is known for its robustness to heteroscedastic noise as shown in [5].

### Figure 2 : Sensitivity analysis

In figure 2, we extend the sensitivity analysis perfomed for spectral clustering (Figure 5 in the submitted paper) to DR scores. This figure shows that tSNEkhorn outperforms tSNE on a wide range of perplexity values. We hope this partly answers reviewer s51e's request about sensitivity analysis.

### Table 1 : Runtime experiments

In table 1, as requested by reviewer 8Btp, we display the runtimes of tSNE and tSNEkhorn on a few datasets. Even though both algorithms have a quadratic complexity with respect to the number of samples, tSNEkhorn's optimization is a bit more involved and requires more time than tSNE.

### Table 2 : Sensibility to PCA dimensions

Finally, to answer the question of reviewer 8Btp, we test different values of dimensions for the pre-processing PCA step. We find that the performance of both tSNE and tSNEkhorn, measured by the silhouette score on COIL/Olivetti/UMNIST, is stable accross these values.


[5] Boris Landa, Ronald R Coifman, and Yuval Kluger. Doubly stochastic normalization of the
gaussian kernel is robust to heteroskedastic noise. SIAM journal on mathematics of data science, 3(1):388–413, 2021.

---

### Decision · Program_Chairs · 2023-09-21

**Decision:**

Accept (poster)

**Comment:**

The paper brings new insights by developing a connection between Entropic Affinities and Optimal Transport. This new connections yields a novel Dimensionality Reduction algorithm whose name is the title of the paper. There was consensus that this is a paper which will be of interest to Neuritis community.